# OBJECT FIDELITY DIFFUSION FOR REMOTE SENSING IMAGE GENERATION

**Ziqi Ye**[1,2]*, **Shuran Ma**[3]*, **Jie Yang**[2], **Xiaoyi Yang**[1], **Yi Yang**[1], **Ziyang Gong**[4],
**Xue Yang**[4][†][‡], **Haipeng Wang**[1][†]

[1]Key Lab for Information Science of Electromagnetic Waves (MoE), Fudan University
[2]Shanghai Innovation Institute    [3]Xidian University
[4]School of Automation and Intelligent Sensing & SCS, Shanghai Jiao Tong University
Code:https://github.com/VisionXLab/OF-Diff

## ABSTRACT

High-fidelity, controllable remote sensing layout-to-image generation is highly valuable for providing high-quality data for downstream object detection tasks. However, existing methods either rely on additional textual guidance, leading to geometric distortions, or require extra real-image references, limiting practical applicability. To address these challenges, we propose Object Fidelity Diffusion (OF-Diff), which leverages object layouts to extract structural shape priors and employs an online-distillation strategy to integrate complex image features. This allows the model to perform highly controllable, high-fidelity image generation at inference without relying on real-image references. Furthermore, we introduce DDPO to fine-tune the diffusion process, making the generated remote sensing images more diverse and semantically consistent. Comprehensive experiments demonstrate that OF-Diff outperforms state-of-the-art methods in the remote sensing across key quality metrics. Notably, the performance of several polymorphic and small object classes shows significant improvement. For instance, the mAP increases by 8.3%, 7.7%, and 4.0% for airplanes, ships, and vehicles, respectively.

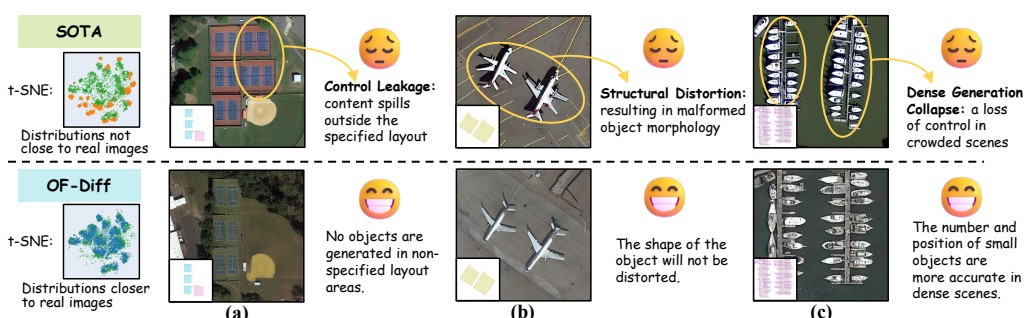

Figure 1: Critical failure modes in the SOTA method (CC-Diff): (a) Control Leakage; (b) Structural Distortion; (c) Dense Generation Collapse. Our OF-Diff (2nd row) effectively resolves these issues.

## 1 INTRODUCTION

Synthesizing high-fidelity, spatially-controllable remote sensing (RS) images is a critical frontier for overcoming the data limitations that hinder downstream perception tasks like object detection (Yang et al., 2021; Zhang et al., 2020; Yang et al., 2019). Current RS generation methods, however, typically rely on either ambiguous text prompts (Khanna et al., 2023; Sebaq & ElHelw, 2024) or auxiliary conditions like semantic maps (Sebaq & ElHelw, 2024; Tang et al., 2024; Gong et al., 2024; Hu et al., 2025; Jia et al., 2025). While visually plausible, such guidance is fundamentally disconnected from the instance-level ground truth, failing to provide the precise control necessary for effective data augmentation.

---

* Equal contribution. † Corresponding author. ‡ Project lead.

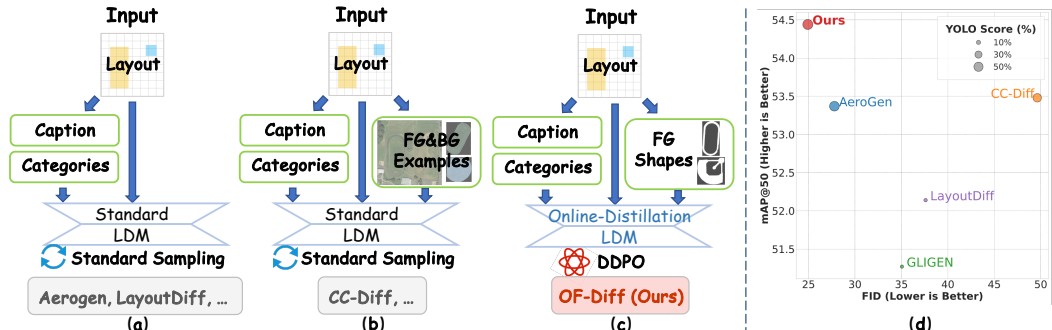

Figure 2: Comparison of OF-Diff with mainstream L2I methods. FG/BG stands for foreground/background. (a) Layout-conditioned baseline. (b) Added instance-based module, limited by quality/quantity of patches from ground truth. (c) OF-Diff enhances fidelity via shape extraction and DDPO, without patch reliance. (d) Results demonstrate superiority.

In contrast, Layout-to-Image (L2I) generation conditioned on object bounding boxes offers a more robust solution for precise spatial control. This paradigm has been extensively studied in the natural image domain—LayoutDiffusion (Zheng et al., 2023) treats it as a multi-modal fusion problem, GLIGEN (Li et al., 2023) enables open-world generation through additional control signals, and ODGen (Zhu et al., 2024) improves controllability by decoupling objects—yet its direct application to remote sensing (RS) imagery remains non-trivial due to expansive backgrounds, arbitrary object orientations, and densely packed scenes.

In RS layout-to-image generation, existing methods like AeroGen (Tang et al., 2025) and CC-Diff (Zhang et al., 2024) take different approaches. AeroGen, a coarse layout-conditioned model, suffers from limited spatial and shape control. In contrast, instance-level methods like CC-Diff achieve higher controllability and fidelity by referencing real instances, but this creates heavy dependence on the quality and quantity of real data, limiting generalization and flexibility.The images generated via CC-Diff diverge more markedly from the real remote sensing data distribution, aligning instead with the style characteristic of the model's pre-training corpus. We summarize common failure modes (see Figure 1), including control leakage, structural distortion, dense generation collapse and feature-level mismatch.

These deficiencies significantly degrade the performance on object detection tasks, limiting their practical application in intelligent RS interpretation. In this paper, we introduce **O**bject **F**idelity **Diff**usion Model (OF-Diff). It is designed to improve the shape fidelity and layout consistency of object generation in RS images. As shown in Figure 2, the existing L2I methods are mainly divided into two categories. The first is layout-conditioned baseline, as shown in Figure 2(a), like Aerogen and LayoutDiffusion. The second is the method with instance-based module, as shown in Figure 2(b), like CC-Diff. However, such methods require real instances and images as references during the sampling stage in order to generate high-quality synthetic images. In contrast, OF-Diff generates high-fidelity remote-sensing objects using only the foreground shape, and subsequently employs online-distillation to further align the outputs with real images, as shown in Figure 2(c). In addition, it fine-tunes the diffusion with DDPO, effectively enhancing the performance of downstream tasks for the generated images. The results in Figure 2(d) demonstrates the superiority of OF-Diff over other methods. Our contributions are summarized as follows:

- We introduce OF-Diff, an online-distillation controllable diffusion model with prior shape extraction, which improves generation fidelity while reducing reliance on real images, enhancing practical applicability.

- We propose a controllable generation pipeline that fine-tunes diffusion models with DDPO for remote sensing images, further boosting fidelity and diversity.

- Extensive experiments demonstrate that OF-Diff generates high-fidelity, layout- and shape-consistent images with dense objects, and serves as an effective enhancement for object detection tasks.

## 2 RELATED WORK

### 2.1 ADVANCES IN IMAGE GENERATION

Diffusion models (Dhariwal & Nichol, 2021; Ho et al., 2020; Kingma et al., 2021) have increasingly replaced Generative Adversarial Networks (GANs) (Goodfellow et al., 2014; Karras et al., 2021) and Variational Autoencoders (VAEs) (Kingma et al., 2013; Rezende et al., 2014) in image synthesis tasks due to their training stability and superior output quality. Recent advances in efficient samplers, such as DDIM (Song et al., 2021), Euler (Karras et al., 2022), and DPM-Solver (Lu et al., 2022), have further improved the practicality. Latent Diffusion Models (LDMs) (Rombach et al., 2022b), which operate in low-dimensional latent spaces, significantly reduce computational costs while preserving visual fidelity. The success of models like DALL·E2 (Ramesh et al., 2022) and Imagen (Saharia et al., 2022) demonstrates how this paradigm supports training on vast internet-scale datasets. As a result, diffusion-based approaches now provide a strong foundation for high-quality image generation.

### 2.2 LAYOUT-TO-IMAGE GENERATION

Controllable image synthesis primarily includes text-to-image (T2I) and layout-to-image (L2I) generation. While T2I models (Nichol et al., 2022; Ramesh et al., 2022) achieve semantic alignment via textual prompts, L2I methods offer better spatial control. Recent works enhance layout conditioning through layout-as-modality designs (Zheng et al., 2023), gated attention (Li et al., 2023), and instance-wise generation (Wang et al., 2024; Zhou et al., 2024). However, these methods rely solely on coarse layout inputs (e.g., bounding boxes), which lack fine-grained shape information critical for synthesizing morphologically complex objects.

### 2.3 REMOTE SENSING IMAGE SYNTHESIS

Synthesizing high-fidelity training data is crucial for advancing remote sensing (RS) object detection, a field critical to numerous applications, but often hampered by the scarcity of extensively annotated datasets. Despite its necessity, most generative models for RS imagery, such as DiffusionSat (Khanna et al., 2023) and RSDiff (Sebaq & ElHelw, 2024), still rely on coarse semantic guidance. While other approaches leverage diverse control signals (Tang et al., 2024) like OpenStreetMaps (Espinosa & Crowley, 2023), they are generally not optimized for the bounding box format central to object detection. This naturally motivates L2I approaches including AeroGen (Tang et al., 2025) and CC-Diff (Zhang et al., 2024), which have improved spatial accuracy and contextual consistency through layout-mask attention and FG/BG dual re-samplers. However, they suffer from limited controllability and heavy reliance on real data.

## 3 METHOD

### 3.1 PRELIMINARY

Diffusion models (Song et al., 2021) aim to capture the underlying data distribution $p(x)$ by iteratively reconstructing data from a noisy representation that is initially sampled from a standard normal distribution. Denoising Diffusion Probabilistic Models (Ho et al., 2020) parameterize the model as the function $\epsilon_\theta(x_t, t)$ to predict the noise component of the sample $x_t$ at any time step $t$. The training objective is to minimize the mean squared error (MSE) loss between the actual noise $\epsilon$ and the predicted noise $\epsilon_\theta(x_t, t)$:

$$\mathcal{L} = \mathbb{E}_{x_t, t, \epsilon \sim \mathcal{N}(\mathbf{0}, \mathbf{I})} \left[ \|\epsilon - \epsilon_\theta(x_t, t)\|^2 \right]. \tag{1}$$

Stable Diffusion (SD) (Rombach et al., 2022b; Qiu et al., 2025) utilizes a pre-trained VQ-VAE Van Den Oord et al. (2017) to encode images into a lower-dimensional latent space, and trains the diffusion model in this latent domain on the representation $z \in Z$. At diffusion step $t$, the training loss under the conditions of text prompt $c_t$ and control conditions $c_c$ can be written as:

$$\mathcal{L} = \mathbb{E}_{z_t, t, c_t, c_c, \epsilon \sim \mathcal{N}(\mathbf{0}, \mathbf{I})} \left[ \|\epsilon - \epsilon_\theta(z_t, t, c_t, c_c)\|^2 \right]. \tag{2}$$

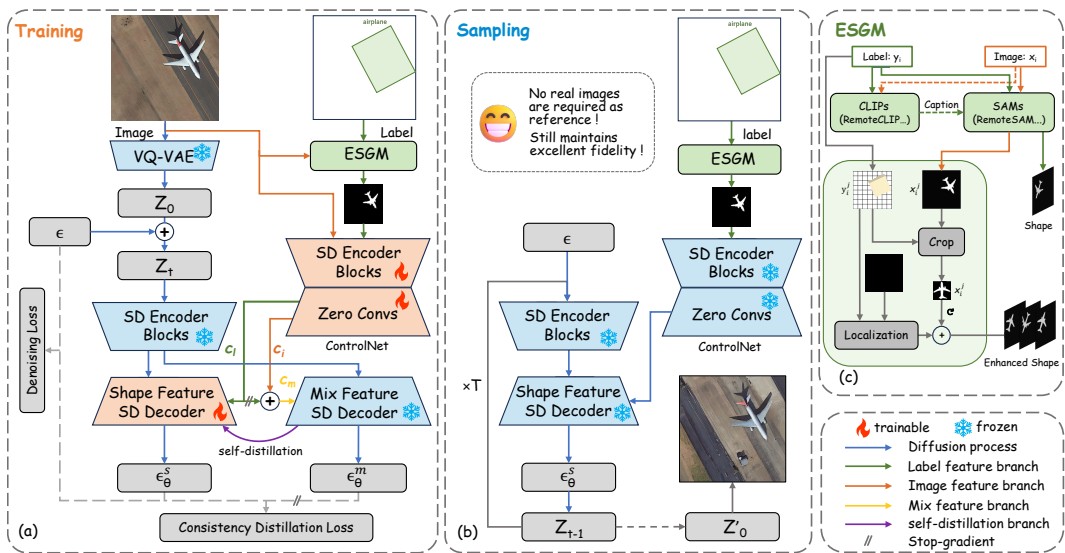

Figure 3: OF-Diff's overall architecture. (a) During training, object shape features extracted by ESGM and image features are processed by ControlNet, and the resulting information is used to update stable diffusion decoders via online-distillation. (b) During sampling, only the label and the shape feature stable diffusion decoder are used to generate synthetic images. (c) Architecture of the Enhanced Shape Generation Module (ESGM).

where $\mathcal{L}$ represents the overall learning objective of the complete diffusion model. This objective function is explicitly applied during the fine-tuning of diffusion models in conjunction with ControlNet (Zhang et al., 2023).

## 3.2 ARCHITECTURE OF OF-DIFF

As illustrated in Figure 3(a), the training of OF-Diff requires both real images and their corresponding labels. **First**, for ControlNet, the real image and its label are processed by the Enhanced Shape Generation Module (ESGM) to extract the object mask. The image and mask are then fed into ControlNet to obtain the image feature $c_i$ and the shape feature $c_s$. To enrich the structural-only shape prior with richer appearance and contextual cues from the image, we combine them into a mix-feature $c_m$, which will later serve as a teacher input in online-distillation. Concretely:

$$c_{\mathrm{m}} = \frac{n}{N} \cdot c_i + \mathrm{sg}\left[c_s\right], \tag{3}$$

where $n$ denotes the current iteration number, and $N$ is the total number of iterations. In order to enable the prediction conditioned on mix-feature to serve as a stable anchor point, to improve the morphological fidelity of the generation, we adopt a stop-gradient strategy (Chen & He, 2021) for $c_s$ when calculating $c_m$.

**Second**, for Stable Diffusion, the input image is first compressed into latent space features $z_0$ by a pre-trained VQ-VAE. Then, it is concatenated with Gaussian noise $\epsilon$ to form $z_t$. After passing through the SD encoder blocks, the feature $Z_t$ is fed into a dual-decoder architecture. One branch, the shape-feature SD decoder, conditions on $c_s$; the other, the mix-feature SD decoder, conditions on $c_m$. Their reconstruction losses are defined as $L_s$ and $L_M$, respectively:

$$\mathcal{L}_s = \mathbb{E}\left[\|\epsilon_\theta^s - \epsilon\|^2\right], \epsilon_\theta^s = \epsilon_\theta\left(z_t, t, c_t, c_s\right), \tag{4}$$

$$\mathcal{L}_m = \mathbb{E}\left[\|\epsilon_\theta^m - \epsilon\|^2\right], \epsilon_\theta^m = \epsilon_\theta\left(z_t, t, c_t, c_m\right), \tag{5}$$

**Third**, for online distillation, the mix-feature SD produces more accurate predictions thanks to its stronger image prior, but needs real images, limiting diversity. In contrast, the shape-feature SD

supports arbitrary label control but risks converging to low-fidelity local minima. To reconcile these trade-offs, we propose an online-distillation framework with a consistency loss $L_c$:

$$\mathcal{L}_c = \mathbb{E}\left[\|\epsilon_\theta^s - \text{sg}\,[\epsilon_{\theta'}^m]\|^2\right]. \tag{6}$$

Here, the prediction $\epsilon_{\theta'}^m$ from mix-feature SD decoder acts as a stop-gradient teacher signal, serving as an anchor to guide the prediction $\epsilon_\theta^s$ from shape-feature SD decoder towards high-fidelity optima in parameter space.

The overall training objective is therefore:

$$\mathcal{L} = \mathcal{L}_s + \mathcal{L}_m + \lambda\mathcal{L}_c, \tag{7}$$

During the sampling phase, as illustrated in Figure 3(b), only the frozen ControlNet and the shape feature stable diffusion are utilized with arbitrary label prior control to synthesize RS images.

## 3.3 Enhanced Shape Generation Module

In natural imagery, perspective and scale changes prevent a unique geometric model for most objects. Conversely, remote-sensing objects display quasi-invariant shapes. For instance, courts are rectangular, chimneys and oil tanks circular, and airplanes bilaterally symmetric with a distinct nose and tail. This shape consistency enables the use of masks to impose strong controllability on image synthesis for remote sensing. To better leverage category labels for object shape extraction, we introduce the Enhanced Shape Generation Module (ESGM, see Figure 3(c)). During the training phase, ESGM uses paired images and labels to generate precise object masks. And at sampling time, it employs learned shape priors to synthesize diverse masks of object shape.

For the given image $x_i$ and its bounding box $y_i^j$ corresponding to category $j$ ($j \in [1, N]$), we first utilize the RemoteCLIP (Liu et al., 2024) to generate a textual description of the object enclosed within the bounding box. With this description and the original image $x_i$, the RemoteSAM (Yao et al., 2025) then generates the corresponding shape masks $\{x_i^j\}$.

In the shape augmentation phase, each object mask $x_i^j$ is cropped by its bounding box $y_i^j$, randomly rotated, and placed back onto a blank canvas to produce a shape-enhanced mask. During training, ESGM uses real image shapes; at sampling, it selects enhanced shapes from a lightweight mask pool collected during or after training. In our experiments, we use masks generated during training.

## 3.4 DDPO Fine-tuning

To enhance the diversity of the distribution of data generated by the fine-tuned model and maintain better consistency with the distribution of real images Schulman et al. (2015; 2017), denoising diffusion policy optimization (DDPO) Black et al. (2023) is applied in the post-training of OF-Diff. DDPO regards the denoising process of the diffusion model as a multi-step Markov decision process (MDP) (for a detailed derivation, please refer to the Appendix A.2). To optimize the policy $\pi(a_t \mid s_t)$ so as to maximize the cumulative reward $\mathbb{E}_{\tau \sim p(\cdot|\pi)!}\left[\sum_{t=0}^{T} R(\mathbf{s}_t, \mathbf{a}_t)\right]$, the gradient $\hat{g}$ is computed as follows:

$$\hat{g} = \mathbb{E}\left[\sum_{t=0}^{T} \frac{p_\theta(\mathbf{x}_{t-1} \mid c,\, t,\, \mathbf{x}_t)}{p_{\theta'}(\mathbf{x}_{t-1} \mid c,\, t,\, \mathbf{x}_t)} \cdot r(\mathbf{x}_0, c) \cdot \nabla_\theta \log p_\theta(\mathbf{x}_{t-1} \mid c,\, t,\, \mathbf{x}_t)\right] \tag{8}$$

$$r(\mathbf{x}_0, c) = \left(KNN(\mathbf{x}_0, \mathbf{x}_0) - \omega KL(\mathbf{x}_0, \mathbf{x}_0')\right) \tag{9}$$

The reward functions based on K-Nearest Neighbor (KNN) and KL divergence are introduced, respectively, to optimize the diversity of generated data and the distribution consistency between generated data and real data. $\omega$ is the weight parameter, and $\mathbf{x}_0'$ is the real image in the dataset. Following standard practice, we compute the KNN in the low-dimensional embedding space of CLIP's image encoder. The implementation details are shown in Appendix A.2.

## 4 EXPERIMENTS

### 4.1 EXPERIMENTAL SETTINGS

**Datasets. DIOR-R** (Cheng et al., 2022), the rotated variant of DIOR (Li et al., 2020), contains 20 categories annotated with oriented bounding boxes; we follow the official 1:1:2 split for training/validation/testing. **DOTA-v1.0** (Xia et al., 2018) includes 15 categories featuring dense scenes and small objects. We crop the images from DOTA to $512 \times 512$ following MMRotate (Zhou et al., 2022), discarding those without valid objects. **HRSC2016** (Liu et al., 2017) is a high-resolution ship detection dataset with a multi-level hierarchical taxonomy. We use the finest-grained level, consisting of 26 detailed ship categories. The experiments of this dataset are reported in Appendix A.4 Unless otherwise specified, we train the diffusion model on the trainset. For downstream detection, we use the trainset annotations as layout and mix generated samples with the real trainset, and report evaluation results on the testset.

**Implementation Details.** We train OF-Diff separately on each dataset (DIOR/DOTA), based on the Stable Diffusion 1.5 (Rombach et al., 2022a) pretrained model. Only the ControlNet and shape feature SD decoder are fine-tuned, while all other modules remain frozen. The weighting coefficient $\lambda$ of the consistency loss is set to 1, the k value in KNN is set to 50, and the weight $\omega$ of the KL divergence is set to 2. Training is performed using the AdamW optimizer with a learning rate of 1e-5. The global batch size is set to 64, and training runs for 100 epochs.

**Benchmark Methods.** We compare our method with state-of-the-art L2I generation models for both remote sensing (AeroGen (Tang et al., 2025), CC-Diff (Zhang et al., 2024)), and natural images (LayoutDiffusion (Zheng et al., 2023), GLIGEN (Li et al., 2023)). For a fair comparison, all models are re-trained using our dataset settings, following their official training details respectively.

**Evaluation Metrics.** To more comprehensively evaluate the effectiveness of OF-Diff, we adopt a total of 13 metrics spanning 4 different evaluation aspects.

- **Generation Fidelity.** We use **FID** (Heusel et al., 2017) and **KID** (Bińkowski et al., 2018) to assess perceptual quality, along with **CMMD** (Jayasumana et al., 2024), which measures CLIP feature distances between generated and real images to evaluate layout alignment.

- **Layout Consistency.** We report **CAS** (Ravuri & Vinyals, 2019) using a pretrained classifier to assess object recognizability, and **YOLOScore** by applying a pretrained Oriented R-CNN (Xie et al., 2021) (w/. Swin Transformer backbone (Liu et al., 2021), MMRotate) to generated images for instance-level consistency.

- **Shape Fidelity.** To assess the geometric quality of generated instances, we perform pairwise comparisons with ground-truth shapes. Each instance pair is cropped, resized to $64 \times 64$, and converted to edge maps. We compute five metrics: **IoU**, **Dice**, Chamfer Distance (**CD**), Hausdorff Distance (**HD**), and **SSIM** (Wang et al., 2004).

- **Downstream Utility.** We train a detector on mixed real and generated images and report $\textbf{mAP}_{50}$, $\textbf{mAP}_{75}$, and overall **mAP** on real test data using Oriented R-CNN (Swin backbone) with a batch size of 24 on 8×NVIDIA 4090 GPUs.

### 4.2 QUALITATIVE RESULTS

**Comparative Results.** Figure 4 compares the generation results of OF-Diff with other methods. OF-Diff not only generates more realistic images but also has the best controllability. For instance, in the first two cases, OF-Diff successfully controlled the number and layout information of the generated objects. The third and fourth cases demonstrate the accuracy of OF-Diff in generating small targets, which other algorithms fail to do accurately. The last case shows the superiority of OF-Diff over other algorithms when generating objects with complex shapes such as airplane.

**Diversity Results.** The images generated by OF-Diff consistently present plausible textures and realistic object shapes, as shown in Figure 6 in Appendix. For instance, airplanes rendered at different orientations maintain coherent semantic relationships with their surrounding environments. Even in small-object scenes (some of which are grayscale remote-sensing images from the DOTA dataset), OF-Diff can still generate visually faithful and geometrically accurate results.

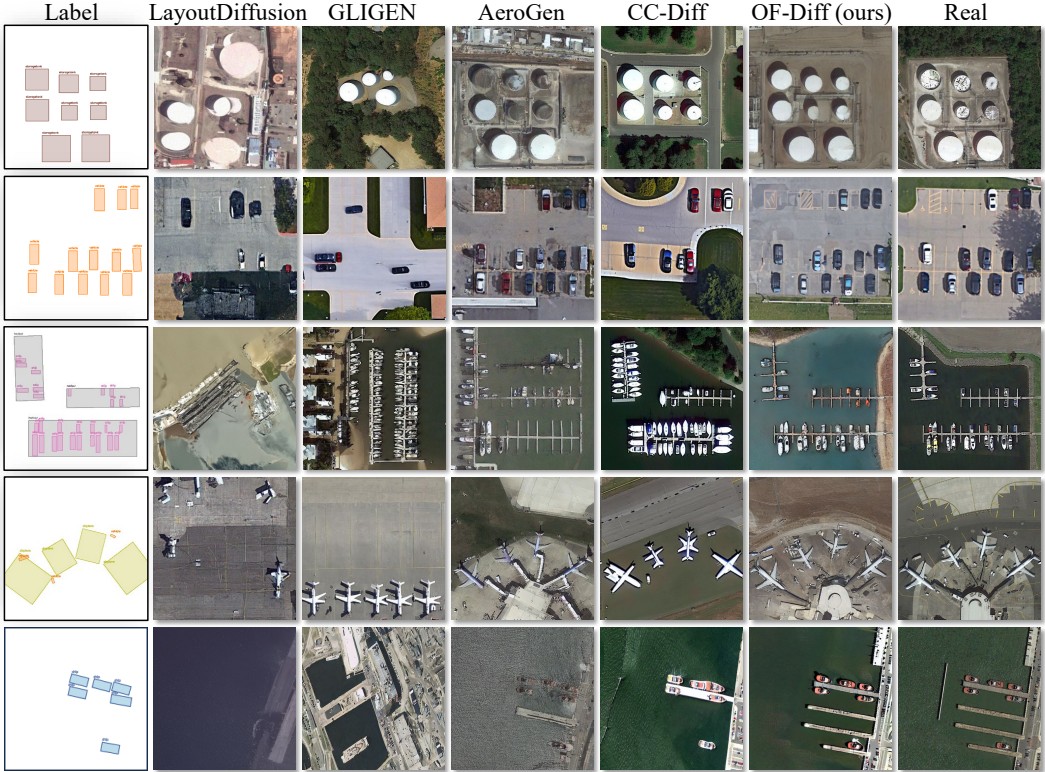

Figure 4: Qualitative results on DIOR, DOTA and HRSC2016. OF-Diff is more realistic and fidelity compared to other methods.

Table 1: Quantitative comparison with SOTA methods on DIOR and DOTA. We evaluate performance on **generation fidelity** (FID, KID, CMMD), **layout consistency** (CAS, YOLOScore) and **trainability** (mAP). OF-Diff demonstrates superior overall performance.

| Method | DIOR Dataset | | | | | | DOTA Dataset | | | | | |
|---|---|---|---|---|---|---|---|---|---|---|---|---|
| | FID↓ | KID↓ | CMMD↓ | CAS↑ | YOLOScore↑ | mAP$_{50}$ | FID↓ | KID↓ | CMMD↓ | CAS↑ | YOLOScore↑ | mAP$_{50}$ |
| LayoutDiff | 37.60 | 0.015 | 0.447 | 70.32 | 7.01 | 52.14 | 21.73 | 0.015 | 0.288 | 77.56 | 21.43 | 66.75 |
| GLIGEN | 35.06 | **0.010** | 0.622 | 76.41 | 6.51 | 51.27 | 39.79 | 0.026 | 0.357 | 76.19 | 15.58 | 66.10 |
| AeroGen | 27.78 | 0.013 | 0.563 | 81.69 | 55.38 | 53.37 | 26.65 | 0.017 | 0.298 | 81.91 | 44.85 | 67.09 |
| CC-Diff | 49.62 | 0.024 | 0.685 | **82.61** | 42.17 | 53.48 | 32.40 | 0.019 | 0.279 | 81.63 | 49.62 | 66.52 |
| **Ours** | **24.92** | 0.011 | **0.312** | 82.55 | **58.99** | **54.44** | **20.84** | 0.014 | **0.271** | **83.79** | **55.68** | **67.89** |

## 4.3 QUANTITATIVE RESULTS

**Generation Fidelity and Consistency.** We compared OF-Diff with state-of-the-art generation methods in remote sensing, including layoutDiffusion (Zheng et al., 2023), GLIGEN (Li et al., 2023), AeroGen (Tang et al., 2025), and CC-Diff (Zhang et al., 2024). The performance of these methods is reported in Table 1. OF-Diff achieved nearly the best performance in both generation fidelity metrics (FID, KID, CMMD) and layout consistency metrics, especially on the DOTA dataset. Additional results are available in the appendix A.4 for the HRSC2016 dataset.

**Trainability of Object Detection.** Following the data enhancement protocol in (Chen et al., 2023), we double the training samples using OF-Diff and assess detection results with the expanded dataset. As shown in Table 9 in Appendix A.7, OF-Diff performs the best on both DIOR and DOTA with mAP improved by 2.2% and 1.94% compared to baseline,respectively. Notably, the performance of several polymorphic and small object classes shows significant improvement. According to Figure 5 (a) and (b), the AP$_{50}$ increases by 8.3%, 7.7%, and 4.0% for airplane, ship, and vehicle on DIOR, and 7.1%, 5.9% and 4.4% for swimming pool, small vehicle, and large vehicle on DOTA.

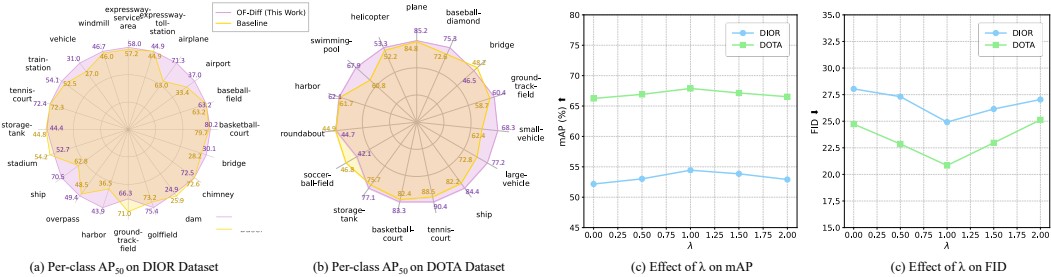

(a) Per-class AP$_{50}$ on DIOR Dataset    (b) Per-class AP$_{50}$ on DOTA Dataset    (c) Effect of $\lambda$ on mAP    (c) Effect of $\lambda$ on FID

Figure 5: AP$_{50}$ on DIOR and DOTA.

Table 2: Object-Shape Fidelity on Canny Edge Maps. We measure the morphological similarity between generated and ground-truth instances by computing IoU, DICE, Chamfer Distance (CD), Hausdorff Distance (HD), and SSIM.

| Method | DIOR Dataset | | | | | DOTA Dataset | | | | |
|---|---|---|---|---|---|---|---|---|---|---|
| | IoU↑ | Dice↑ | CD↓ | HD↓ | SSIM↑ | IoU↑ | Dice↑ | CD↓ | HD↓ | SSIM↑ |
| LayoutDiff | 0.0497 | 0.0908 | 12.037 | 25.962 | 0.1667 | 0.0402 | 0.0748 | 15.229 | 30.202 | 0.2194 |
| GLIGEN | 0.055 | 0.1002 | 12.257 | 25.850 | 0.1652 | 0.0645 | 0.1182 | 10.432 | 23.196 | 0.1967 |
| AeroGen | 0.0855 | 0.153 | 8.209 | 20.314 | 0.2142 | 0.0863 | 0.1536 | 8.1386 | 20.687 | 0.2261 |
| CC-Diff | 0.0891 | 0.1582 | 8.0909 | 20.066 | 0.1963 | 0.0692 | 0.1255 | 9.6226 | 21.247 | 0.2171 |
| **Ours** | **0.1009** | **0.1763** | **7.6579** | **19.459** | **0.2691** | **0.1205** | **0.2045** | **6.6317** | **17.311** | **0.2938** |

**Object-Shape Fidelity.** We measure the morphological similarity between the generated instances and ground truth by calculating the Intersection over Union (IoU), DICE coefficient, Chamfer distance (CD), Hausdorff distance (HD), and Structural Similarity Index (SSIM), based on the Canny Edge Map. As shown in Table 2, the results demonstrate that OF-Diff attains state-of-the-art performance in all evaluation metrics for object-shape fidelity. Specifically, we first convert the rotated bounding box (R-Box) to a horizontal bounding box (H-Box) and crop the instance with a 20% padding to ensure the full object is captured. The cropped patches are then resized to $64 \times 64$ pixels, and their shapes are extracted using cv2.Canny. For a detailed qualitative comparison, Figure 11 in Appendix A.8 visualizes the instance patches and their corresponding edge maps from different methods. Each image set is ordered as follows: Ground Truth, OF-Diff, AeroGen, CC-Diff, GLIGEN, and LayoutDiff, demonstrating our method's superior ability to adhere to object shapes.

**Adaptability of Unknown Layout.** To evaluate robustness of these methods, we also generate images based on the unknown layouts during the training phase. According to Table 3, for unknown layout, OF-Diff performs well in terms of generation fidelity, layout consistency, and trainability. In downstream tasks, OF-Diff still delivers a 1.54% mAP gain over the second-best method.

**The Detailed Results of Downstream.** Table 10 and 11 in Appendix A.8 report the average precision (AP) obtained by the competing generative methods over multiple categories in the downstream tasks. From Tables 10, it can be observed that OF-Diff (ours) achieves a clear advantage in several categories. For instance, OF-Diff achieves superior performance on Airplane (71.3%), Golf Field (75.4%) and Ship (70.5%), with improvements of approximately 5% to 10% over the second-best method. For a few other categories, OF-Diff does not deliver the top AP, yet the gap to the best result remains marginal. Table 11 shows that, on the DOTA dataset, OF-Diff obtains the highest AP in roughly half of all categories and still delivers notable gains in categories such as Small-vehicle (68.3%), Ship (84.4%) and Swimming-pool (67.9%).

## 4.4 ABLATION STUDY

We assessed the impact of different modules on image generation semantic consistency and downstream trainability by OF-Diff in Table 4. We found that the images generated with captions are more in line with semantic consistency and human aesthetics, but the fidelity of these images decreases. This is equivalent to the data distribution deviating from the real dataset and being more inclined towards the data distribution during pre-training. We conduct human/GPT assessments and

Table 3: Quantitative comparison on the unknown layout dataset during training (DIOR Val).

| Method | Unknown Layout during Training | | | | | | | |
| | FID↓ | KID↓ | CMMD↓ | CAS↑ | YOLO Score↑ | mAP↑ | mAP$_{50}$↑ | mAP$_{75}$↑ |
|---|---|---|---|---|---|---|---|---|
| LayoutDiff | 44.58 | 0.018 | 0.539 | 29.34 | 10.37 | 30.41 | 53.07 | 32.07 |
| GLIGEN | 39.56 | 0.013 | 0.444 | 66.36 | 2.13 | 30.06 | 52.68 | 31.29 |
| AeroGen | 28.62 | 0.013 | 0.276 | 80.78 | 46.36 | 32.98 | 55.11 | 34.26 |
| CC-Diff | 49.92 | 0.024 | 0.513 | 78.01 | **51.74** | 32.49 | 53.72 | 35.39 |
| **Ours** | **24.18** | **0.012** | **0.271** | **83.34** | 49.59 | **33.02** | **56.65** | **36.17** |

Table 4: Ablation study: impact of ESGM, Online-distillation $L_c$, and DDPO on semantic consistency (CAS) and downstream trainability (YOLOScore and mAP$_{50}$).

| ESGM | $L_c$ | DDPO | FID ↓ | KID ↓ | CMMD ↓ | CAS ↑ | YOLOScore ↑ | mAP$_{50}$ ↑ |
|---|---|---|---|---|---|---|---|---|
| ✗ | ✗ | ✗ | 42.59 | 0.029 | 0.965 | 80.27 | 41.20 | 52.13 |
| ✓ | ✗ | ✗ | 24.87 | 0.012 | 0.428 | 82.16 | 55.08 | 52.76 |
| ✗ | ✓ | ✗ | 36.25 | 0.021 | 0.596 | 81.57 | 46.27 | 53.14 |
| ✗ | ✗ | ✓ | 41.26 | 0.027 | 0.815 | 81.06 | 42.53 | 53.41 |
| ✓ | ✓ | ✗ | 24.98 | **0.010** | 0.313 | 82.30 | 57.83 | 54.31 |
| ✓ | ✗ | ✓ | 25.78 | 0.013 | 0.368 | 82.37 | 58.26 | 54.17 |
| ✗ | ✓ | ✓ | 37.98 | 0.025 | 0.613 | 81.91 | 47.74 | 53.21 |
| ✓ | ✓ | ✓ | **24.92** | 0.011 | **0.312** | **82.55** | **58.99** | **54.44** |

a fine-grained feature analysis in Appendix A.7, which collectively reveal the nature of this trade-off. Therefore, the ablation experiments for each module were conducted based on the absence of caption input. The contribution of each module to the enhancement of image generation fidelity is evaluated by incorporating additional components into the diffusion model with online-distillation. DDPO indicates whether to fine-tune the trained diffusion model through reinforcement learning. Results show that Enhanced Shape Generation Module (ESGM), Online-Distillation ($L_c$) and the DDPO based on KNN and KL Divergence effectively improve the performance metrics. Notably, ESGM can substantially improve the YOLOScore by over 10%. In addition, we vary the weighting coefficient $\lambda$ in the consistency loss (Eq. 7) to assess its impact on mAP and FID. As shown in Figure 5 (c) and (d), both metrics are optimal at $\lambda = 1$.

## 4.5 DISCUSSION

As shown in Figure 10 in Appendix A.3, the inclusion of additional captions as input has a significant impact on the outcomes of image generation. Specifically, incorporating captions enhances the aesthetic appeal of the generated images, resulting in richer and more visually pleasing color compositions. However, this improvement comes at a cost: similar to CC-Diff, it leads to a deviation of the generated data distribution from that of the original real data. In contrast, when no additional captions are provided as input, although the generated images may appear less aesthetically refined, their data distribution remains closer to that of real images. A user study from both human and GPT-5 in Table 8 have confirmed this. Further analysis of the generated-image distribution and the impact of aesthetics on performance is provided in the Appendix A.7.

## 5 CONCLUSION

Existing image generation methods struggle to precisely generate dense small objects and those with complex shapes, such as numerous small vehicles and airplanes in remote sensing images. To address this, we introduce OF-Diff, an online-distillation controllable diffusion model with prior shapes extraction and DDPO. During the training phase, we extract the prior shape of objects to enhance controllability and use a online-distillation diffusion with parameter sharing to improve the model's learning ability for real images. Therefore, in the sampling phase, OF-Diff can generate images with high fidelity without real images as references. Finally, we fine-tune the diffusion by DDPO that combines KNN and KL divergence to make the synthesized images more realistic and consistent. Extensive experiments demonstrate the effectiveness and superiority of OF-Diff in generating small and difficult objects with complex structures and dense scenes in remote sensing.

ACKNOWLEDGEMENT

This work was supported in part by the National Natural Science Foundation of China under Grants 62271153 and 62506229, the Natural Science Foundation of Shanghai under Grant 25ZR1402268, and the Shanghai QiYuan Innovation Foundation.

ETHICS STATEMENT

This work adheres to the ICLR Code of Ethics. Our study focuses on generating synthetic remote sensing images using diffusion models, and does not involve human subjects or personally identifiable information. All datasets used are publicly available and appropriately licensed for research purposes. We have taken care to avoid introducing harmful biases or misrepresentations in the generated images. While OF-Diff aims to improve object fidelity and controllability in synthetic data for downstream tasks such as object detection, we acknowledge that misuse of generated images could have unintended consequences. Researchers using this technology should ensure ethical and lawful application, and consider potential societal impacts. No conflicts of interest or undisclosed sponsorship influenced this work.

REPRODUCIBILITY STATEMENT

We have made efforts to ensure the reproducibility of our results. The main paper provides detailed descriptions of the OF-Diff architecture, online-distillation training strategy, and DDPO fine-tuning process. All datasets used are publicly available, and the data preprocessing and layout-to-image preparation steps are described in the main text and Appendix. Hyperparameters, training schedules, and evaluation metrics are fully documented in the supplementary materials. Additionally, we provide anonymized source code and configuration files to facilitate replication of our experiments.

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

## A    APPENDIX

### A.1    THE USE OF LARGE LANGUAGE MODELS (LLMS)

Large Language Models (LLMs) were used in this work exclusively for language polishing and improving the clarity of writing. No LLM was used for generating scientific ideas, experimental design, or data analysis. All technical content, results, and interpretations presented in this paper are solely the work of the authors. The authors take full responsibility for the content of this paper, including any text refined with the assistance of LLMs.

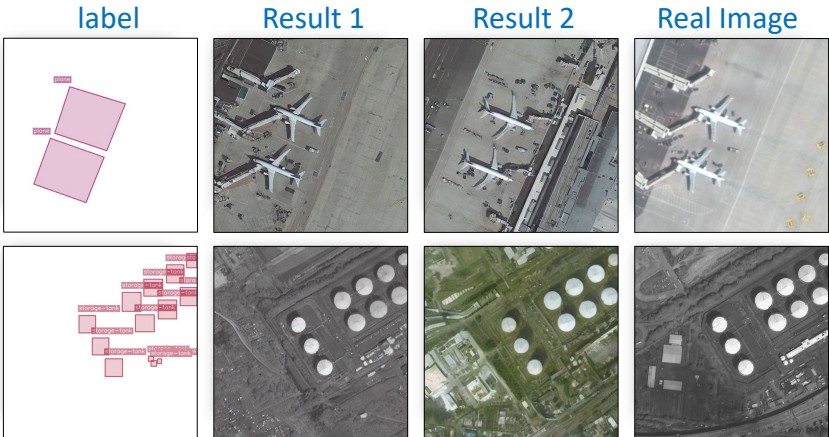

Figure 6: The diversity of different results from the same OF-Diff model.

## A.2 REINFORCEMENT LEARNING STRATEGY

The mapping relationship is defined as follows:

$$\pi(a_t \mid s_t) \triangleq p_\theta(\mathbf{x}_{t-1} \mid \mathbf{x}_t, c) \tag{10}$$

$$P(s_{t+1} \mid s_t, a_t) \triangleq (\delta_c, \delta_{t-1}, \delta_{\mathbf{x}_{t-1}}) \tag{11}$$

$$\rho_0(s_0) \triangleq \big(p(c), \delta_T, \mathcal{N}(0, I)\big) \tag{12}$$

$$R(s_t, a_t) \triangleq \begin{cases} r(\mathbf{x}_0, c), & \text{if } t = 0, \\ 0, & \text{otherwise.} \end{cases} \tag{13}$$

where $\delta_z$ denotes the Dirac delta distribution whose probability density is zero everywhere except at $z$. The symbols $s_t$ and $a_t$ represent the state and action at time $t$, respectively. Specifically, $s_t$ is defined as the tuple composed of the condition $c$, the time step $t$, and the noisy image $\mathbf{x}_t$ at that time, whereas $a_t$ is defined as the noisy image $\mathbf{x}_{t-1}$ from the preceding time step. The policy is denoted by $\pi(a_t \mid s_t)$, the transition kernel by $P(s_{t+1} \mid s_t, a_t)$, the initial state distribution by $\rho_0(s_0)$, and the reward function by $R(s_t, a_t)$.

For detailed DDPO policy, we employ a ResNet101 pre-trained on ImageNet-1K as our feature extraction model, and utilize KNN and KL divergence to compute both the diversity among generated images and their similarity to real images. Let $X$ denote the set of generated images, $Y$ represent the real images, where $x_i \in X$, $y_j \in Y$, and $M$ is our feature extraction model.

The KNN reward is calculated as follows: 1) First, we extract features from $X$ using model $M$: $F_x = M(X)$. 2) For each feature vector $f_x^i \in F_x$, we compute its K-nearest neighbors among all feature vectors $f_x^j \in F_x$. The KNN reward for $x_i$ is the average of these K nearest neighbor distances, denoted as $KNN(f_x^i, F_x)$. In our implementation, we set K to 50.

The KL reward is calculated as follows: 1) We extract features from both $X$ and $Y$ using model $M$: $F_x = M(X)$ and $F_y = M(Y)$. 2) For each feature vector $f_x^i \in F_x$ and $f_y^j \in F_y$, we compute $KL(f_x^i, f_y^i)$ for each $i$, and use $-KL(f_x^i, f_y^i)$ as the KL reward for $x_i$.

In summary, the reward for a generated image $x_i$ is computed as:

$$r_x^i \triangleq KNN(f_x^i, F_x) - wKL(f_x^i, f_y^i) \tag{14}$$

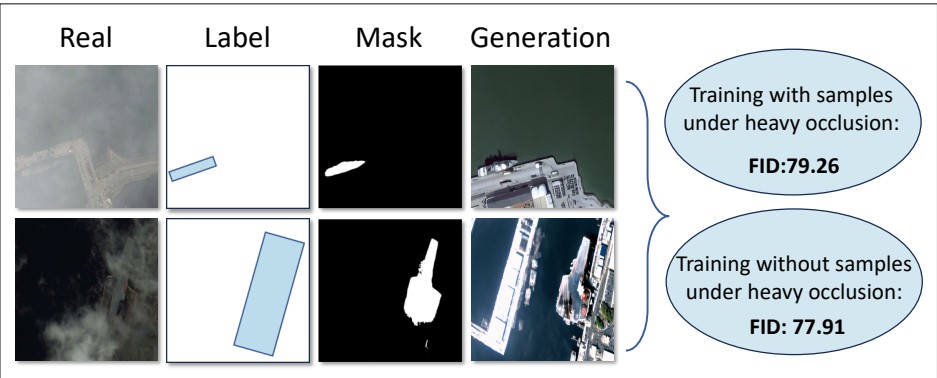

Figure 7: The results of OF-Diff in handling objects under heavy occlusion. The results indicate that while severe occlusion does indeed cause a certain degree of degradation in the quality of the target mask extracted by ESGM, it has little impact on the generated quality FID.

### A.3 ANALYSIS AND DISCUSSION

According to the current experimental results, adding the DDPO strategy does not simultaneously outperform previous results on all metrics. Using reinforcement learning strategies can indeed improve the performance of downstream tasks, but it does not necessarily improve the quality of image generation simultaneously. In other words, reinforcement learning strategies can also purposefully improve the quality of image generation, but this may come at the cost of not improving the performance of downstream tasks.

The proposed OF-Diff injects object shape masks extracted from the image layout as controllable conditions into the diffusion model, which effectively enhances object fidelity and improves the generation of small objects. However, this also makes the model dependent on the quality of the extracted shape masks. We analyze the impact of a distorted mask on the model's generated results. Specifically, we selected cases such as objects under heavy occlusion to examine the model's generation performance. Based on the analysis results in Figure7, we found that even under severe occlusion conditions, ESGM still demonstrates strong object mask extraction and generation capabilities. However, when the generated mask shape exhibits certain anomalies, it does produce objects matching that distorted shape. Nevertheless, this does not affect the overall FID and trainability of the generated images. Although the shapes we currently extract may exhibit edge anomalies in the object mask due to occlusion and other issues, complete errors are extremely rare.

### A.4 QUANTITATIVE RESULTS ON HRSC2016 DATASET

Table 5 reports the comparative results on HRSC2016, where our method consistently achieves strong performance. Although it ranks second on CMMD, CAS, and YOLOScore—which mainly reflect aesthetic quality or local recognizability—it attains the best results on FID and KID, which measure distribution fidelity, as well as on the most crucial downstream metric, mAP50, outperforming the second-best method by +1.5%. This indicates that our generated data preserves the real remote-sensing distribution more faithfully and thus provides more effective support for downstream tasks. A more detailed analysis is provided in Appendix A.7.

Table 5: Fidelity and Downstream Performance on HRSC2016

| Method | HRSC216 Dataset | | | | | |
|---|---|---|---|---|---|---|
| | FID↓ | KID↓ | CMMD↓ | CAS↑ | YOLOScore↑ | mAP$_{50}$ ↑ |
| LayoutDiff | 120.68 | 0.152 | 1.763 | 24.51 | 2.51 | 56.97 |
| GLIGEN | 92.92 | 0.037 | 0.634 | 35.41 | 5.03 | 39.72 |
| AeroGen | 97.44 | 0.055 | **0.51** | 39.62 | 16.4 | 47.68 |
| CC-Diff | 84.55 | 0.035 | 0.681 | **45.27** | **32.42** | 62.57 |
| **Ours** | **77.91** | **0.026** | 0.573 | 42.19 | 30.97 | **64.1** |

Table 6: The impact of real and generated images at different ratios on mAP for downstream tasks.

| Data Composition | mAP (%) |
|---|---|
| 100% Generated | 45.67 (-7.17) |
| 50% Real + 50% Generated | 50.74 (-2.10) |
| 100% Real | 52.84 |
| 100% Real + 50% Generated | 53.92 (+1.08) |
| 100% Real + 100% Generated | 54.38 (+1.54) |
| 100% Real + 200% Generated | 54.74 (+1.90) |
| 100% Real + 300% Generated | 54.82 (+1.98) |

Table 7: The data on the computational cost of training OF-Diff compared to the key baselines.

| Models | Train GPU Mean Memory (MB) | Train GPU Hours | Inference Mean Time/Sample (s) |
|---|---|---|---|
| LayoutDiff | 29232 | 41.33 | <1s |
| GLIGEN | 14186 | 57.76 | 5.18 |
| AeroGen | 27634 | 49.52 | 1.85 |
| CC-Diff | 13668 | 38.01 | 3.96 |
| OF-Diff | 27340 | 44.27 | 3.42 |

### A.5 THE mAP EVOLUTION GIVEN DIFFERENT AMOUNTS OF SYNTHETIC AND REAL DATA

We conduct multiple experiments on trainability using different quantities of real and generated data. The results are shown in the Table 6. Experimental results indicate that using only 100% synthetic data struggles to achieve downstream task performance comparable to real data. However, this also demonstrates that even without a single real image, relying solely on synthetic images can enable object detection algorithms to achieve a mAP of 45.67%. Furthermore, training with a larger volume of generated images can effectively enhance the model's object detection capabilities. However, when the amount of generated data reaches three times that of real data (based on the generation setting described in the paper), downstream performance shows little further improvement.

### A.6 THE COMPUTATIONAL COST

We provide the data on the computational cost of training OF-Diff compared to the key baselines in the Table .7. Experimental results indicate that although OF-Diff is not the most optimal in terms of training costs (GPU memory and GPU hours) and inference time, it remains nearly the second-best among these methods and does not incur high computational costs.

### A.7 AESTHETIC−DOWNSTREAM PERFORMANCE CONFLICT

To further reveal the potential conflict between aesthetic quality and downstream task performance, we conduct a three-part analysis consisting of questionnaire evaluation, downstream performance comparison, and feature-level visualization.

(1) **Human/GPT questionnaire study.** As shown in Fig. 8, we design two targeted questions:

Q1. Which image more closely matches the style of real remote-sensing imagery? (e.g., realistic noise patterns, texture details, natural illumination, authentic object boundaries)

Q2. Which image looks more aesthetically pleasing? (e.g., clarity, color harmony, contrast, smoothness, visual comfort, overall appearance)

We invite 8 PhD researchers and 8 remote-sensing experts to participate, and additionally perform 3 rounds of GPT-5 evaluation. For each class in DIOR, we randomly sample one pair of images generated with vs. without captions (from the same ground truth), resulting in 20 image pairs. Each pair is randomly shuffled to avoid positional bias. The results are shown in Table 8, each

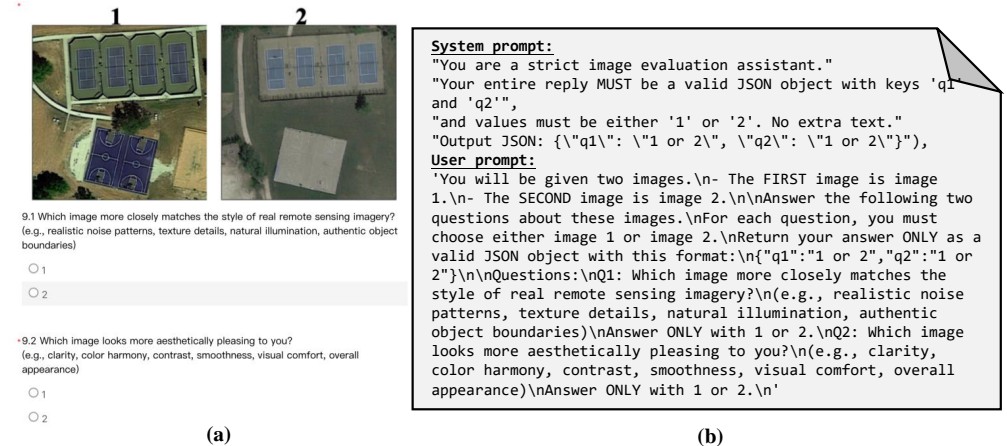

Figure 8: Aesthetic evaluation questionnaire design for generated images: (a) human experts, (b) GPT-5.

value represents the average frequency with which the corresponding option was selected across all questionnaires.

Table 8: Single-choice results from human experts and GPT-5 (averaged over multiple annotators or repeated evaluations).

| Option | Human experts | | GPT-5 | |
|---|---|---|---|---|
| | Q1 | Q2 | Q1 | Q2 |
| w./ caption | 6.57 | **11.21** | 2.33 | **15.33** |
| w./o. caption | **13.43** | 8.79 | **17.67** | 4.67 |

Both human experts and GPT consistently prefer the caption-conditioned images in terms of aesthetics, but find them less similar to real remote-sensing imagery. In contrast, images generated without captions appear less visually appealing but better preserve the real-world texture and structural characteristics needed for downstream tasks.

**(2) Downstream performance comparison.** On the DIOR dataset, as shown in the table at the bottom-right of Fig. 9, adding captions reduces the downstream improvement $\Delta\text{mAP}_{50}$ by 1.15 and also leads to a significantly higher FID. Combined with finding (1), this reveals that caption-guided generation tends to over-beautify images—masking the natural imperfections of remote-sensing imagery—and consequently harms downstream performance.

**(3) Feature-level visualization.** We also visualize features using t-SNE in Fig. 9. We observe that adding captions produces more outliers, whereas samples generated without captions align more closely with the GT distribution, indicating higher fidelity.

**Taken together**, these findings suggest that models should remain faithful to the inherent quirks and imperfections of the original remote-sensing data, rather than generating overly "idealized" or aesthetically enhanced imagery. Incorporating captions risks amplifying the latter behavior.

Additional examples are provided in Fig. 10.

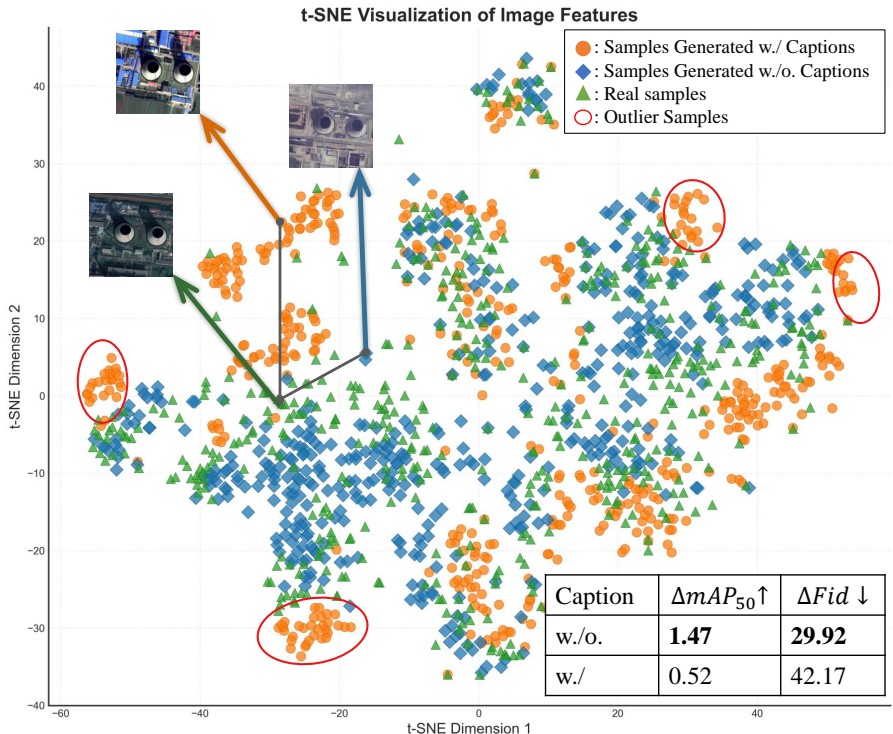

Figure 9: t-SNE feature analysis of generated samples w./ and w./o. captions. Incorporating captions produces a large number of outlier samples, lowers fidelity (higher FID), and further degrades downstream performance (lower $\Delta mAP_{50}$).

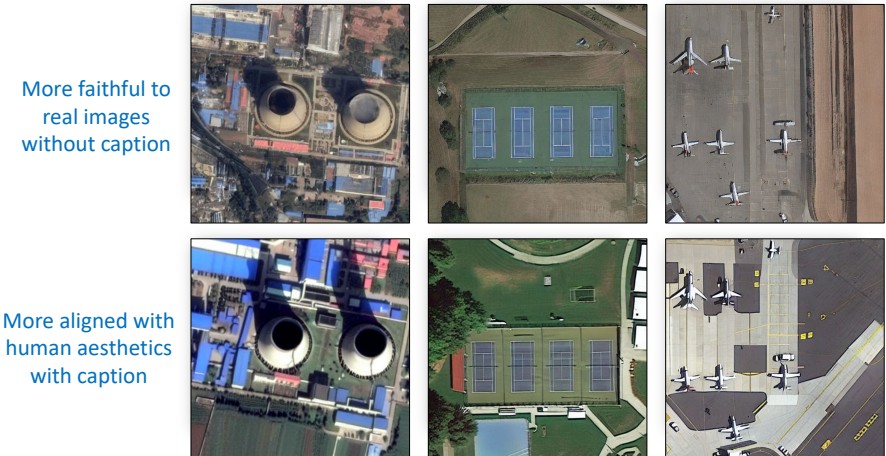

Figure 10: The influence of caption on the generation of images in terms of being more realistic and more aesthetically pleasing.

## A.8    MORE QUALITATIVE AND QUANTITATIVE RESULTS

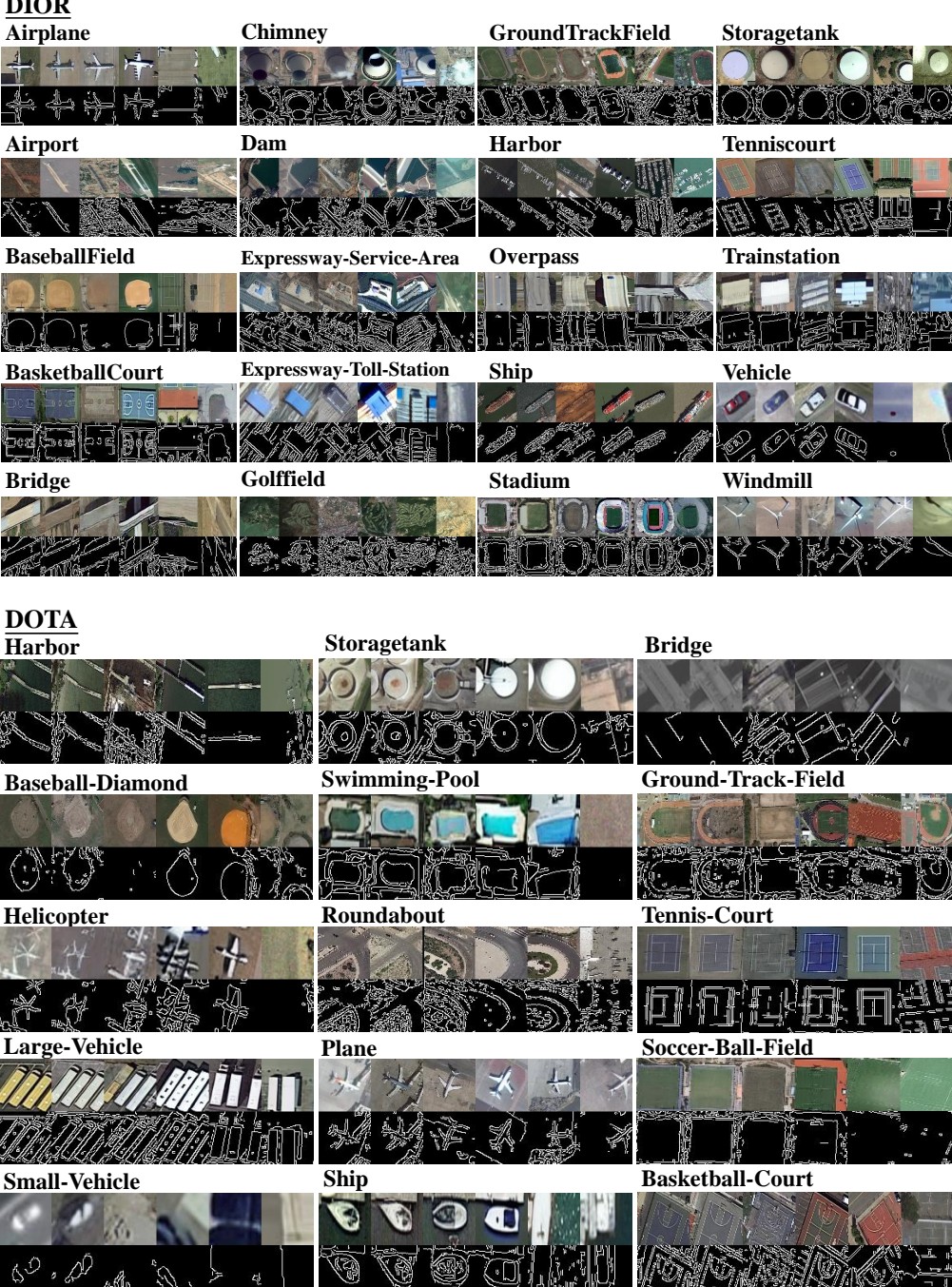

Figure 11: Comparison of generated instance patches and their Canny edge maps for the same bbox on the DIOR and DOTA dataset. Each image set is ordered as follows: Ground Truth, OF-Diff, AeroGen, CC-Diff, GLIGEN, and LayoutDiff.

Table 9: Trainability (↑) comparison on DIOR and DOTA. 'Baseline' denotes accuracy with the unaugmented dataset.

| Method | DIOR Dataset | | | DOTA Dataset | | |
|---|---|---|---|---|---|---|
| | mAP | $mAP_{50}$ | $mAP_{75}$ | mAP | $mAP_{50}$ | $mAP_{75}$ |
| Baseline | 30.51 | 52.84 | 32.10 | 38.09 | 66.31 | 38.44 |
| LayoutDiff | 29.81 | 52.14 | 30.36 | 38.91 | 66.75 | 40.37 |
| GLIGEN | 28.48 | 51.27 | 29.21 | 38.84 | 66.10 | 40.24 |
| AeroGen | 31.53 | 53.37 | 33.60 | 38.45 | 67.09 | 39.07 |
| CC-Diff | 31.82 | 53.48 | 33.97 | 38.51 | 66.52 | 39.02 |
| **Ours** | **32.71** | **54.44** | **34.05** | **40.03** | **67.89** | **42.20** |

Table 10: Detailed downstream trainability results (measured by average precision) on the DIOR dataset.

| Method | DIOR Dataset | | | | | | | | | |
|---|---|---|---|---|---|---|---|---|---|---|
| | Expressway Service-area | Expressway toll-station | Airplane | Airport | Baseball field | Basketball court | Bridge | Chimney | Dam | Golf Field |
| LayoutDiff | 53.1 | 44.8 | 62.8 | 29.4 | 63.2 | 79.6 | 25.9 | 72.6 | 22.4 | 69.3 |
| GLIGEN | 52.7 | 44.8 | 62.6 | 26.7 | 63.0 | 79.6 | 25.2 | 72.6 | 19.5 | 67.4 |
| AeroGen | **58.1** | **45.2** | 63.1 | 32.7 | **63.4** | **81.0** | 29.5 | 72.6 | 21.1 | 69.1 |
| CC-Diff | 53.5 | 45.1 | 62.9 | **38.4** | 63.3 | 79.9 | 29.3 | **72.7** | **27.6** | 70.5 |
| **Ours** | 58.0 | 44.9 | **71.3** | 37.0 | 63.2 | 80.2 | **30.1** | 72.5 | 24.9 | **75.4** |

| Method | DIOR Dataset | | | | | | | | | |
|---|---|---|---|---|---|---|---|---|---|---|
| | Ground Track-field | Harbor | Overpass | Ship | Stadium | Storage Tank | Tennis Court | Trainstation | Vehicle | Windmill |
| LayoutDiff | **71.2** | 32.8 | 43.9 | 62.9 | 59.0 | **52.5** | 72.4 | 52.1 | 26.9 | 46.0 |
| GLIGEN | 70.1 | 30.3 | 45.8 | 62.8 | 56.8 | 52.0 | **72.5** | 49.0 | 26.9 | 45.3 |
| AeroGen | 71.0 | 42.7 | **50.7** | 62.9 | 56.6 | 44.5 | **72.5** | 52.6 | **31.4** | **46.7** |
| CC-Diff | 64.6 | 43.1 | 49.0 | 63.0 | **61.7** | 44.7 | 72.4 | **54.4** | 27.0 | 46.5 |
| **Ours** | 66.3 | **43.9** | 49.4 | **70.5** | 52.7 | 44.4 | 72.4 | 54.1 | 31.0 | **46.7** |

Table 11: Detailed downstream trainability results (measured by average precision) on the DOTA dataset.

| Method | DOTA Dataset | | | | | | | |
|---|---|---|---|---|---|---|---|---|
| | Plane | Baseball-diamond | Bridge | Ground Track-field | Small-vehicle | Large-vehicle | Ship | Tennis-court |
| LayoutDiff | 80.4 | 74.2 | **48.8** | 59.9 | 62.9 | 72.7 | 82.5 | 89.6 |
| GLIGEN | 87.0 | 72.9 | 47.3 | 56.4 | 63.7 | 73.1 | 82.7 | 90.1 |
| AeroGen | 86.1 | **77.3** | 48.6 | 58.6 | 64.5 | **78.1** | 82.5 | 83.3 |
| CC-Diff | **87.2** | 73.4 | 47.1 | 57.8 | 64.3 | 73.9 | 82.6 | 89.2 |
| **Ours** | 85.2 | 75.3 | 46.5 | **60.4** | **68.3** | 77.2 | **84.4** | **90.4** |

| Method | DOTA Dataset | | | | | | |
|---|---|---|---|---|---|---|---|
| | Basketball-court | Storage-tank | Soccer-ball-field | Roundabout | Harbor | Swimming-pool | Helicopter |
| LayoutDiff | 78.9 | 76.3 | **46.3** | **47.6** | 60.9 | 62.1 | **57.9** |
| GLIGEN | 79.6 | 76.5 | 42.2 | 43.1 | 60.7 | 62.3 | 53.8 |
| AeroGen | 77.3 | 79.9 | 44.8 | 46.6 | 59.4 | 62.6 | 56.4 |
| CC-Diff | 79.0 | **82.7** | 42.7 | 43.1 | 58.9 | 62.7 | 52.9 |
| **Ours** | **83.3** | 77.1 | 42.1 | 44.7 | **62.1** | **67.9** | 53.3 |

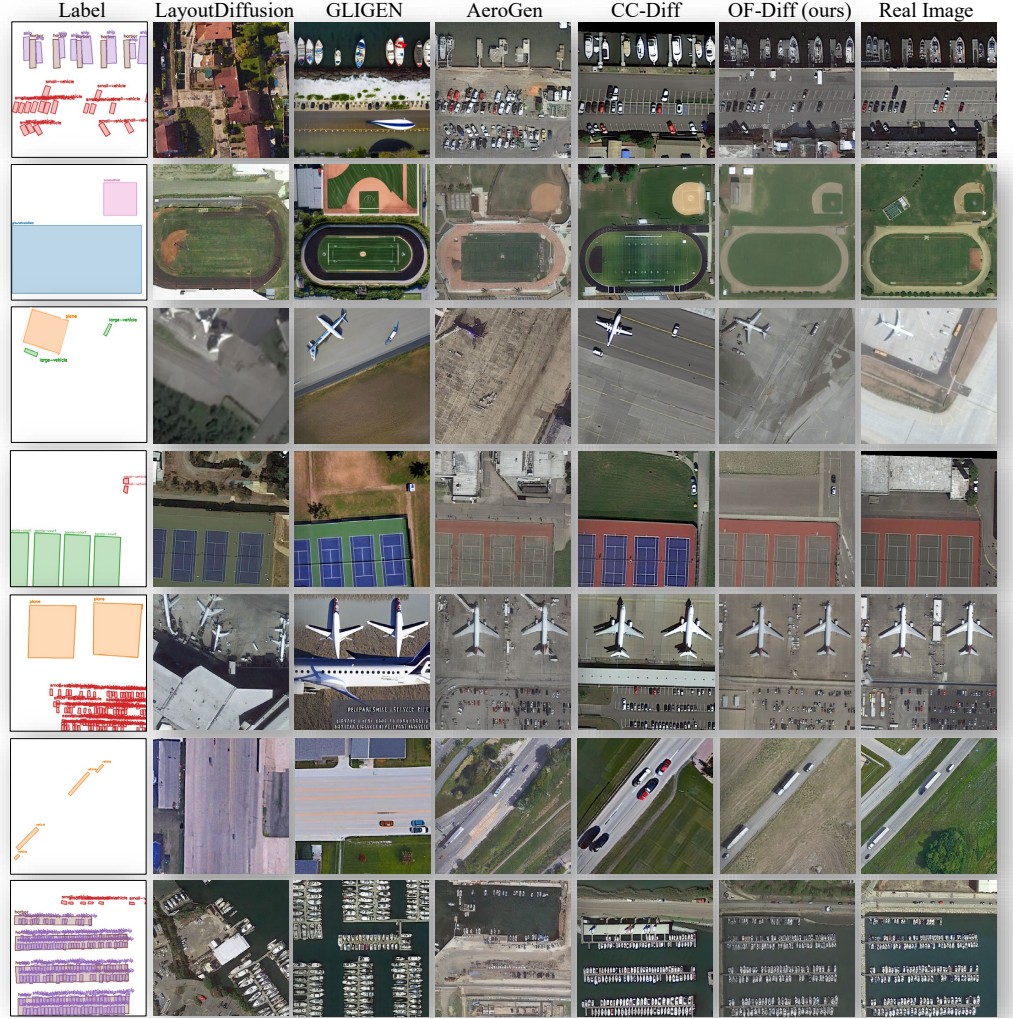

Figure 12: Additional qualitative results on DIOR and DOTA. The results demonstrate that OF-Diff has certain superiority and accuracy in generating small objects, and it also has an advantage in generating the shapes of objects. For instance, the aircraft target in the third row is generated more accurately by OF-Diff, with a more realistic structure. The small vehicles in the fourth and fifth rows and the large vehicle in the sixth row are also more accurately generated. Additionally, the small ship in the seventh row is generated with greater accuracy.

