# OpenReview forum: "Object Fidelity Diffusion for Remote Sensing Image Generation"
_ICLR.cc/2026/Conference — ICLR 2026 Poster_

### Official Review · Reviewer_7tpr · 2025-10-26

**Soundness:** 2
**Presentation:** 2
**Contribution:** 2
**Rating:** 4
**Confidence:** 5

**Summary:**

This paper proposes an object fidelity diffusion method, which extends the layout-to-image paradigm into the remote sensing sense. This method combines many existing modules to achieve controllable image generation.

**Strengths:**

1. This is a layout-to-image generation method for the remote sensing field.
2. The proposed method integrates an online-distillation strategy and DDPO fine-tuning to achieve high-fidelity image generation.

**Weaknesses:**

1. The proposed OF-Diff is primarily an assembly of existing, well-known components: ControlNet for conditioning, a form of online distillation for feature alignment, and DDPO for fine-tuning.
2. The central claim of "reducing reliance on real images" is misleading; the method heavily depends on real images during training to extract shape masks via RemoteSAM and to train the teacher model in the distillation process. The "prior shape extraction" is essentially semantic segmentation, which is a standard technique. The paper fails to convincingly demonstrate what the key, novel insight is, beyond a specific combination of these existing blocks.
3. The comparison methods discussed in the paper (such as LayoutDiffusion and GLIGEN) were not specifically designed for remote sensing imagery. Applying them directly to remote sensing scenarios and using them as baselines is unfair. Furthermore, the authors did not provide sufficiently detailed training configurations (such as whether all comparison methods underwent adequate adaptation to remote sensing data), which may have led to biased comparison results.
4. Although the paper demonstrates improvements in mAP for object detection tasks, it does not explicitly prove whether these gains stem from enhanced image quality or merely increased data volume. There is a lack of quantitative analysis regarding the actual contribution of generated images during detector training, such as through visualization or feature distribution alignment analysis.
5. The entire pipeline is critically dependent on the quality of masks produced by the ESGM, which itself relies on external models (RemoteCLIP, RemoteSAM). The paper provides no analysis of what happens when this module fails or produces noisy/incorrect masks, which is inevitable in practice. How robust is OF-Diff to errors in the initial shape prior? If the extracted mask is distorted, will the generation process fail catastrophically? This is a major point of practical vulnerability that is completely unaddressed.
6. The proposed system is exceptionally complex, involving multiple stages: ESGM, a dual-branch diffusion model with online distillation, and a subsequent DDPO fine-tuning step. The computational cost, memory footprint, and training time must be enormous compared to baselines like AeroGen. The paper completely omits any discussion of efficiency, training time, or inference speed. The practical utility is questionable if the method is an order of magnitude more expensive to train and deploy.

**Questions:**

1. Can you precisely state the novel algorithmic contribution of OF-Diff, distinct from the existing components (ControlNet, distillation, DDPO) it builds upon?
2. Have you attempted to fine-tune the CC-Diff baseline on your specific datasets to ensure a fair comparison? Could its distribution shift be mitigated?
3. How does the performance of OF-Diff degrade when the input shape masks from ESGM are noisy or partially incorrect? Please provide a robustness analysis.
4. Can you provide data on the computational cost (e.g., GPU hours, memory usage) of training OF-Diff compared to the key baselines?

---

> ### Author Response · Authors · 2025-11-21
> **Official Comment by Authors**
>
> Thank you very much for taking the time to review and for your support. We try our best to address your questions as follows.
>
> > **Weakness1 and Question1: The proposed OF-Diff is primarily an assembly of existing, well-known components: ControlNet for conditioning, a form of online distillation for feature alignment, and DDPO for fine-tuning. Can you precisely state the novel algorithmic contribution of OF-Diff, distinct from the existing components (ControlNet, distillation, DDPO) it builds upon?**
>
> Thank you for this important concern. OF-Diff is not a module, but a systematic, implementable framework.
>
> The innovation of OF-Diff lies not in individual components, but in proposing a novel end-to-end training framework that for the first time organizes ControlNet, online distillation, and DDPO into a closed-loop optimization process for continuously calibrating the generation distribution toward downstream detection tasks. Additionally, we designed task-specific structural-consistency and detector-friendly rewards, which enable the generator to progressively converge toward the distribution of real detection data, a capability not achieved by prior work.
>
> > **Weakness2: The central claim of "reducing reliance on real images" is misleading; the method heavily depends on real images during training to extract shape masks via RemoteSAM and to train the teacher model in the distillation process. The "prior shape extraction" is essentially semantic segmentation, which is a standard technique. The paper fails to convincingly demonstrate what the key, novel insight is, beyond a specific combination of these existing blocks.**
>
> Thank you for this important concern.
>
> In other image generation methods, real images are all required during training. (See: [1] Li, Y., Liu, H., Wu, Q., Mu, F., Yang, J., Gao, J., ... & Lee, Y. J. (2023). Gligen: Open-set grounded text-to-image generation. In Proceedings of the IEEE/CVF conference on computer vision and pattern recognition (pp. 22511-22521). [2] Zhou, D., Li, Y., Ma, F., Zhang, X., & Yang, Y. (2024). Migc: Multi-instance generation controller for text-to-image synthesis. In Proceedings of the IEEE/CVF conference on computer vision and pattern recognition (pp. 6818-6828). The advantage of OF-Diff lies in its ability to perform inference without requiring any real image data, offering a significant improvement over the previous state-of-the-art method CC-Diff (which requires real object reference images and background reference images). Our approach effectively leverages the shape characteristics of objects in remote sensing images, such as cylindrical oil tanks and rectangular basketball courts. This improves object fidelity in the generated images. Additionally, this achieves high fidelity during inference solely through prior object shape masks, without requiring real data.
> > **Weakness3 and Question2: The comparison methods discussed in the paper (such as LayoutDiffusion and GLIGEN) were not specifically designed for** **remote sensing** **imagery. Applying them directly to remote sensing scenarios and using them as baselines is unfair. Furthermore, the authors did not provide sufficiently detailed training configurations (such as whether all comparison methods underwent adequate adaptation to remote sensing data), which may have led to biased comparison results. Have you attempted to fine-tune the CC-Diff baseline on your specific datasets to ensure a fair comparison? Could its distribution shift be mitigated?**
>
> Thank you for the comment. To ensure fair comparison, we fine-tune all models on each dataset (DIOR-R/DOTA/HRSC2016) using the optimal settings (batchsize/learning rate/optimizer) reported in their papers.
>
> Additionally, for models not specifically developed for remote-sensing imagery, we performed minimal yet necessary adaptations to enable fair evaluation and stable optimization:
>
> - LayoutDiffusion: extended fine-tuning epochs (100 → 250) to ensure convergence on remote-sensing data.
> - GLIGEN: abandon the alpha_type schedule so that control signals are applied at *every* diffusion step, avoiding degraded layout guidance.
> - GLIGEN / LayoutDiffusion: minimally adapted layout encoders (e.g., dimension adjustment) to support polygon annotations, which are used consistently across all compared methods.
>
> These adjustments only ensure proper functionality in the remote-sensing setting and do not introduce unfair advantages.
>
> These are our detailed training parameters:
>
> | method          | epoch | Global batchsize | Learning rate |
> | --------------- | ----- | ---------------- | ------------- |
> | aerogen         | 100   | 4                | 1e-5          |
> | ccdiff          | 100   | 320              | 1e-4          |
> | GLIGEN          | 100   | 32               | 5e-5          |
> | Layoutdiffusion | 250   | 32               | 1e-4          |

---

> ### Author Response · Authors · 2025-11-21
> **Official Comment by Authors**
>
> > **Weakness4: Although the paper demonstrates improvements in mAP for object detection tasks, it does not explicitly prove whether these gains stem from enhanced image quality or merely increased data volume. There is a lack of** **quantitative analysis** **regarding the actual contribution of generated images during detector training, such as through visualization or feature distribution alignment analysis.**
>
> Thank you for this important concern. We primarily explore the impact of generated data on improving mAP for object detection from two aspects: 1. The quality of image generation. 2. The influence of the quantity of generated data, which can be referenced based on Reviewer 8kyc's suggestion.
>
> First, we evaluate the quality of generated images using trainability. After feature extraction from images (with or without captions) that better align with human preferences and real-world scenarios, we visualize them in a two-dimensional space using t-SNE dimensionality reduction. We then compare FID and mAP, as shown in Figure 9 in Appendix A.7. Experimental results indicate that generated images closer to the real data distribution exhibit superior trainability, even when these images do not align with human preferences. Additionally, we conducted weight comparison tests for KNN and KL in reinforcement learning. The results are as follows:
>
> | ω    | Trainability |
> | ---- | ------------ |
> | 0.25 | -7.83%       |
> | 0.5  | -2.74%       |
> | 1    | +0.83%       |
> | 2    | +1.47%       |
> | 4    | +0.51%       |
>
> The results demonstrate that improving image quality can effectively enhance the performance of downstream tasks. However, it is important to note the following: 1. Generated images should exhibit a certain degree of diversity without deviating excessively from real images, as this helps improve the trainability of the data. 2. If images conform too closely to human aesthetic preferences while diverging from the data distribution of real remote sensing images, trainability will also decrease.
>
> Second, we analyzed the volume of generated experimental data, with results shown in Appendix A.5 and Table 6. This demonstrates that training with a larger volume of generated images can effectively enhance the model's object detection capabilities. However, when the amount of generated data reaches three times that of real data (based on the generation setting described in the paper), downstream performance shows little further improvement.

---

> ### Author Response · Authors · 2025-11-21
> **Official Comment by Authors**
>
> > **Weakness5 and Question3: The entire pipeline is critically dependent on the quality of masks produced by the ESGM, which itself relies on external models (RemoteCLIP, RemoteSAM). The paper provides no analysis of what happens when this module fails or produces noisy/incorrect masks, which is inevitable in practice. How robust is OF-Diff to errors in the initial shape prior? If the extracted mask is distorted, will the generation process fail catastrophically? This is a major point of practical** **vulnerability** **that is completely unaddressed. How does the performance of OF-Diff degrade when the input shape masks from ESGM are noisy or partially incorrect? Please provide a robustness analysis.**
>
> Thank you for this important concern. We acknowledge that the model may inherit potential biases from these foundational models. We assess the model's robustness by analyzing the results generated from shape masks of varying quality.
>
> First, we discuss the computational cost. When aiming to improve downstream task performance, RemoteCLIP is unnecessary. We investigated the fidelity, aesthetic quality, and trainability of model-generated images with and without captions; detailed procedures and analysis are provided in the Appendix A.7. Experiments demonstrate that while images without captions are perceived as less aesthetically pleasing by humans, they yield higher performance gains in downstream tasks. Their FID value of 29.92 is significantly better than 42.17. Additionally, RemoteSAM's computational cost is acceptable. We tested improved methods like SAM and RSPrompter. Experiments confirm RemoteSAM holds significant advantages in inference speed and mask generation, with a generation speed of 0.3 seconds per mask being entirely acceptable.
>
> Second, we analyze the impact of a distorted mask on the model's generated results. Specifically, we selected cases such as objects under heavy occlusion to examine the model's generation performance. Based on the analysis results in Appendix A.3 and Figure 7, we found that even under severe occlusion conditions, ESGM still demonstrates strong object mask extraction and generation capabilities. However, when the generated mask shape exhibits certain anomalies, it does produce objects matching that distorted shape. Nevertheless, this does not affect the overall FID and trainability of the generated images. Although the shapes we currently extract may exhibit edge anomalies in the object mask due to occlusion and other issues, complete errors are extremely rare. Nevertheless, we can explore solutions for addressing erroneous shapes in future research.
>
> > **Weakness6 and Question4: The proposed system is exceptionally complex, involving multiple stages: ESGM, a dual-branch diffusion model with online distillation, and a subsequent DDPO fine-tuning step. The computational cost, memory footprint, and training time must be enormous compared to baselines like AeroGen. The paper completely omits any discussion of efficiency, training time, or inference speed. The practical utility is questionable if the method is an order of magnitude more expensive to train and deploy. Can you provide data on the computational cost (e.g.,** **GPU** **hours, memory usage) of training OF-Diff compared to the key baselines?**
>
> Thank you for this important concern. We have provided the data on the computational cost of training OF-Diff compared to the key baselines in the Appendix A.6.
>
> |   models   |  FID  | train GPU mean memory | train GPU hours | inference meantime/sample (s) |
> | :--------: | :---: | :-------------------: | :-------------: | :---------------------------: |
> | Layoutdiff | 37.60 |         29232         |      41.33      |             ＜1s              |
> |   GLEGEN   | 35.06 |         14186         |      57.76      |             5.18              |
> |  aerogen   | 27.78 |         27634         |      49.52      |             1.85              |
> |  cc-diff   | 49.62 |         13668         |      38.01      |             3.96              |
> |  of-diff   | 24.92 |         27340         |      44.27      |             3.42              |
>
> Experimental results indicate that although OF-Diff is not the most optimal in terms of training costs (GPU memory and GPU hours) and inference time, it remains nearly the second-best among these methods and does not incur high computational costs.

---

> > ### Comment · Reviewer_7tpr · 2025-11-27
> >
> > Thank you for your detailed response. After carefully reviewing your rebuttal, most of my concerns are addressed. However, I still have the following unresolved questions:
> >
> > 1. Regarding A2, I understand your point that the capability to "utilize the shape characteristics of objects in remote sensing images to avoid the need for real object reference images and background reference images" is provided by the inherent properties of remote sensing images. However, I do not seem to have observed what specific methodology the authors designed to implement this; it appears this functionality is simply attributed to the image being a remote sensing image itself.
> >
> > 2. I still have concerns about the novelty of this paper. Overall, it primarily combines existing technologies, though it does include some technical contributions. I cannot fully confirm whether this contribution meets the acceptance criteria for the ICLR remote sensing domain. This may require further judgment and decision from the AC and the PC.

---

> ### Author Response · Authors · 2025-11-27
> **Official Comment by Authors**
>
> Dear Reviewer 7tpr,
>
> Thank you for your questions. We are responding to your questions and hope to gain your approval.
>
> > **1. Response to Question 1:**
>
> The feasibility of this approach is supported by the unique properties of object shapes found in remote sensing imagery. We have proposed some methods that rely only on the object's shape characteristics, thus eliminating the need for real object reference images and background reference images.
>
> - First, we designed the ESGM module. Beyond just the model it invokes, ESGM also establishes a high-quality object shape pool. This ensures that complete and diverse object shapes are available for subsequent image generation.
> - Second, to effectively leverage these object shape characteristics from remote sensing images, we introduced a dual-decoder architecture. Additionally, we incorporated online-distillation to help the images generated by shape-feature SD (only shape characteristics) better approximate real images.
> - Third, our method is able to surpass other existing techniques, including the reference image-dependent CC-Diff, on the Object-Shape Fidelity metric (morphological similarity), even without utilizing real targets as a reference.
>
>
> > **2. Response to Question 2:**
>
> (1) We believe this paper is not merely an integration of existing technical solutions; we propose three technical innovations that effectively solve the problem:
>
> - First, we propose ESGM, which provides accurate and effective control conditions for the diffusion model's training process by introducing multi-object prior shapes from the images. Furthermore, it generates high-quality masks, thereby ensuring the diversity and realism of the generated targets.
> - Second, we propose the online-distillation framework, which is based on mix-feature SD and shape-feature SD. We incorporated a consistency loss into this framework. This allows the model to effectively learn real-image object information during training, while still being able to complete high-fidelity generation tasks during inference without requiring real images.
> - Third, we propose a DDPO-based method, which includes our self-designed reward function combining KNN and KL divergence. This enables the model to further enhance the quality and trainability of the generated images.
>
> (2) We argue that a method capable of solving a specific, concrete problem constitutes a significant technical innovation. The approach presented in this paper effectively resolves the distortion problem for generated objects in remote sensing images. For example, generated aircraft images reliably exhibit the correct structure of one nose, two wings, and one tail, as demonstrated in Figure 1(b). Additionally, OF-Diff effectively improves the accuracy of object generation location and quantity in complex scenes, as shown in the third row of Figure 4. This capability was not achievable by previous methods. Furthermore, Reviewer 3Som, HT6t, and 8kyc appear to hold a positive view regarding the innovativeness of this paper. Therefore, we believe that OF-Diff possesses significant technical contributions to the remote sensing domain and represents an innovative piece of work.
>
> Best regards,
>
> Authors

---

> > ### Comment · Reviewer_7tpr · 2025-11-28
> >
> > Thank you for your reply. l have no additional problem.
> > However, I found that the score cannot modified.

---

> ### Author Response · Authors · 2025-11-28
> **Official Comment by Authors**
>
> Dear Reviewer 7tpr,
>
> Thank you for your prompt response. We are pleased that our answers have addressed your problems.
>
> Although the score cannot be modified at this time, we greatly appreciate your constructive approach throughout this rebuttal and discussion. We also thank you for indicating that you would modify the score after we addressed all your questions.
>
> We hope the outcome of this rebuttal and discussion will be considered by the AC and PC, even if the score remains unchanged.
>
> Best regards,
>
> Authors

---

### Official Review · Reviewer_8kyc · 2025-10-31

**Soundness:** 2
**Presentation:** 2
**Contribution:** 2
**Rating:** 6
**Confidence:** 3

**Summary:**

This work proposes a a method for generating satellite images conditioned on object layouts. Motivation for this work is the need of augmenting existing training datasets, thus, proposing a method for generating additional label-image pairs. Furthermore, this work focuses specifically on instance/object level generations (instead of the commonly adopted semantic maps conditioning or text conditioning).

**Strengths:**

- Authors identify a lack in current literature: few works sucessfully tackle instance-level generation given the difficulty of the task. Instance-level (layout-to-image paradigm) gives more precise control over the generations and alignment with the ground truth conditions.
- Authors propose a realiable pipeline for achieving high layout fidelity generations by DDPO finetuning, and without the need of using real control images.
- Authors provide ablation studies for the design decisions.
- Extensive evaluation is carried out.
- Qualitative results look strong compared to other models.

**Weaknesses:**

Authors do not provide any dataset augmentation experiment for OOD-datasets. Such experiment would be useful to prove the usefulness of the model beyond their training dataset distribution, to see if their generations are actually useful for other downstream datasets. I believe this is an important experiment that should be carried out, as it determines the overall usefulness of the generated images not just within the training distribution.

I suggest authors to select some other dataset (not DOTA or DIOR) and compare a baseline model trained on the original dataset and an augmented version of the target dataset.

In fact, Tables 6 and 7 show very small downstream improvements in terms of trainability when compared with other methods. Could authors provide some intuition why generations are not always profitable for training? Trainability is an important part of the work carried out. Authors could provide more ablation experiments showing whether the generated images are usefulness for training.

**Questions:**

- Given a baseline model trained on DIOR/DOTA default dataset, authors show downstream improvements when baseline model is trained with original + OF-Diff generations (Figure 5). It would be interesting to see the AP evolution given different amounts of synthetic vs real data. For instance:
   - Train a baseline model on 100% real images
   - Train a baseline model on 100% generated images
   - Train a baseline model on 50% real + 50% generated images
   - etc.
   - Train a baseline model on 100% real + 50% generated images
   - Train a baseline model on 100% real + 100% generated images
   - Train a baseline model on 100% real + 200% generated images
   - etc.
- Do authors have any intuition when the baseline model performance plateaus? In other words, the point at which generating more images will not improve downstream performance?

---

> ### Author Response · Authors · 2025-11-21
> **Official Comment by Authors**
>
> Thank you very much for taking the time to review and for your support. We try our best to address your questions as follows.
>
> > **Weakness1: Authors do not provide any dataset augmentation experiment for OOD-datasets. Such experiment would be useful to prove the usefulness of the model beyond their training dataset distribution, to see if their generations are actually useful for other downstream datasets. I believe this is an important experiment that should be carried out, as it determines the overall usefulness of the generated images not just within the training distribution.**
>
> Thank you for this valuable suggestion. Following your suggestion, we utilized caption prompts to generate samples with feature distributions distinct from the real data. As shown in Figure 9 in the Appendix A.7, samples marked with red circles indicate those with distributions differing from the real data. Experimental results demonstrate that these samples also enhance detection performance in downstream tasks (+0.52% on mAP). However, based on existing results, out-of-distribution data does not appear to be more effective than data closer to the true image distribution in terms of trainability. Additionally, we will conduct a more extensive and systematic evaluation of data-augmentation strategies on out-of-distribution datasets, which constitutes a substantial future effort.
>
> > **Weakness2: I suggest authors to select some other dataset (not DOTA or DIOR) and compare a baseline model trained on the original dataset and an augmented version of the target dataset.**
>
> Thank you for this valuable suggestion. We employed the additional HRSC-2016 dataset for model performance evaluation to validate its effectiveness in remote sensing applications. This dataset was selected due to its representativeness in remote sensing object detection and its adoption as a benchmark by other methods ([1] Tang, D., Cao, X., Wu, X., Li, J., Yao, J., Bai, X., ... & Meng, D. (2025). AeroGen: Enhancing remote sensing object detection with diffusion-driven data generation. In Proceedings of the Computer Vision and Pattern Recognition Conference (pp. 3614-3624).). We summarize the experimental results for both the original dataset and the augmented version, along with comparisons against other methods in Appendix A.7.
>
> > **Weakness3: In fact, Tables 6 and 7 show very small downstream improvements in terms of trainability when compared with other methods. Could authors provide some** **intuition** **why generations are not always profitable for training? Trainability is an important part of the work carried out. Authors could provide more ablation experiments showing whether the generated images are** **usefulness** **for training.**
>
> Thank you for this valuable suggestion. We explore the impact of generated images on trainability from two perspectives.
>
> First, we investigate the effects of similarity and diversity between generated and real images. Specifically, we measure diversity and similarity using KNN and KL divergence in reinforcement learning. The experimental results are shown in the table below. The experiments indicate that when ω is less than 1, meaning that diversity dominates, the generated images may deviate too far from real images. This severely impairs trainability. When ω is greater than 1, meaning similarity dominates, generated images exhibit good trainability. However, as ω continues to increase, trainability actually weakens. This may occur because the generated images become too similar to real images, preventing downstream task models from learning effective features from this data.
>
> | ω    | Trainability |
> | ---- | ------------ |
> | 0.25 | -7.83%       |
> | 0.5  | -2.74%       |
> | 1    | +0.83%       |
> | 2    | +1.47%       |
> | 4    | +0.51%       |
>
> Second, we explored how generating images that favor human visual preferences versus real-world scenes impacts trainability. Experimental results are presented in the Appendix A.7. After feature extraction, we visualized images (with or without captions) that align more closely with human preferences or real-world scenes using t-SNE dimensionality reduction in a two-dimensional space. This allows for a more intuitive understanding of their respective characteristics.
>
> In summary, we draw two conclusions: 1. Generated images require a certain level of diversity but should not deviate excessively from real images, as this enhances data trainability. 2. If images align too closely with human aesthetics while diverging from the distribution of real remote sensing images, trainability also decreases.

---

> ### Author Response · Authors · 2025-11-21
> **Official Comment by Authors**
>
> > **Question1 & Question2: Given a baseline model trained on DIOR/DOTA default dataset, authors show downstream improvements when baseline model is trained with original + OF-Diff generations (Figure 5). It would be interesting to see the AP evolution given different amounts of synthetic vs real data. For instance: Train on 100% real** **images，train** **on 100% generated images, train on 100% generated images, train on 50% real + 50% generated images, train on 100% real + 50% generated images, train on 100% real + 100% generated images, etc. Do authors have any intuition when the baseline model performance plateaus? In other words, the point at which generating more images will not improve downstream performance?**
>
> Thank you for this valuable suggestion. Following your suggestion, we conduct multiple experiments on trainability using different quantities of real and generated data. The results are shown in the table below and are also included in the Appendix A.5 of the paper.
>
> | Data Composition           | mAP (%)       |
> | -------------------------- | ------------- |
> | 100% Generated             | 45.67 (-7.17) |
> | 50% Real + 50% Generated   | 50.74 (-2.10) |
> | 100% Real                  | 52.84         |
> | 100% Real + 50% Generated  | 53.92 (+1.08) |
> | 100% Real + 100% Generated | 54.38 (+1.54) |
> | 100% Real + 200% Generated | 54.74 (+1.90) |
> | 100% Real + 300% Generated | 54.82 (+1.98) |
>
> Experimental results indicate that using only 100% synthetic data struggles to achieve downstream task performance comparable to real data. However, this also demonstrates that even without a single real image, relying solely on synthetic images can enable object detection algorithms to achieve a mAP of 45.67%. Furthermore, training with a larger volume of generated images can effectively enhance the model's object detection capabilities. However, when the amount of generated data reaches three times that of real data (based on the generation setting described in the paper), downstream performance shows little further improvement.

---

### Official Review · Reviewer_HT6t · 2025-11-01

**Soundness:** 2
**Presentation:** 3
**Contribution:** 3
**Rating:** 6
**Confidence:** 2

**Summary:**

This paper proposes Object Fidelity Diffusion (OF-Diff), a novel diffusion-based model for layout-to-image generation in remote sensing (RS). The key idea is to improve object fidelity and layout controllability without relying on real-image references at inference time. The authors introduce:
1. Enhanced Shape Generation Module (ESGM) to extract object shape priors from bounding box layouts.
2. Online distillation to align shape-based generation with real-image features during training.
3. DDPO (Denoising Diffusion Policy Optimization) to fine-tune the model for better diversity and semantic consistency.
The model is evaluated on DIOR-R and DOTA datasets using a comprehensive set of metrics including FID, KID, YOLOScore, and downstream detection mAP. Results show superior fidelity, layout consistency, and downstream utility, especially for small and polymorphic objects.

**Strengths:**

1. OF-Diff does not require real-image references at inference, a significant practical improvement.
2. State-of-the-art results on both DIOR-R and DOTA datasets, with mAP improvements of up to 8.3% on airplane and 7.7% on ship categories.
3. The paper is well-structured, with clear problem motivation, method description, and experimental analysis.

**Weaknesses:**

1. The online distillation and DDPO fine-tuning steps are computationally expensive, but the paper does not report training time, GPU usage, or memory overhead.
2. The paper shows that adding captions improves aesthetics but hurts fidelity (Fig. 7). However, this trade-off is not deeply analyzed. A user study or perceptual evaluation would help clarify when and why to use captions.
3. The method heavily relies on ESGM-generated shape masks. While the paper mentions that distorted masks lead to poor generation, it does not quantify how robust the model is to noisy or incomplete masks?
4.The model is only evaluated on two datasets (DIOR-R and DOTA), both of which are airborne/satellite optical imagery.

**Questions:**

see weaknesses.

---

> ### Author Response · Authors · 2025-11-21
> **Official Comment by Authors**
>
> Thank you very much for taking the time to review and for your support. We try our best to address your questions as follows.
>
> > **Weakness1: The online distillation and DDPO fine-tuning steps are computationally expensive, but the paper does not report training time,** **GPU** **usage, or memory overhead.**
>
> Thank you for this valuable suggestion. We have provided details on the training time, GPU usage and other information of OF-Diff compared to other methods in the Appendix A.6.
>
> |   models   |  FID  | train GPU mean memory | train GPU hours | inference meantime/sample (s) |
> | :--------: | :---: | :-------------------: | :-------------: | :---------------------------: |
> | Layoutdiff | 37.60 |         29232         |      41.33      |             ＜1s              |
> |   GLEGEN   | 35.06 |         14186         |      57.76      |             5.18              |
> |  aerogen   | 27.78 |         27634         |      49.52      |             1.85              |
> |  cc-diff   | 49.62 |         13668         |      38.01      |             3.96              |
> |  of-diff   | 24.92 |         27340         |      44.27      |             3.42              |
>
> > **Weakness2: The paper shows that adding captions improves aesthetics but hurts fidelity (Fig. 7). However, this trade-off is not deeply analyzed. A user study or perceptual evaluation would help clarify when and why to use captions.**
>
> Thank you for this valuable suggestion.
>
> Following your suggestion, we conducted a user study. We surveyed 14 human remote sensing/AI experts to evaluate two types of generated images based on two criteria: “more consistent with real-world remote sensing imagery” and “more visually appealing to humans.” Details are provided in the Appendix A.7. This study indicates that both human experts and GPT-5 perceive generated images with captions as more aesthetically pleasing, while those without captions appear more realistic.
>
> Additionally, we performed feature extraction on both generated images (with/without captions) and real images, then visualized them in a two-dimensional space using t-SNE dimensionality reduction. This allows for a more intuitive understanding of their respective characteristics. See Figure 9 in the Appendix A.7 for details.
>
> > **Weakness3: The method heavily relies on ESGM-generated shape masks. While the paper mentions that distorted masks lead to poor generation, it does not quantify how** **robust** **the model is to noisy or incomplete masks?**
>
> Thank you for this valuable suggestion. We analyzed the impact of distorted or incomplete masks on the model's generated results. Specifically, we selected cases such as objects under heavy occlusion to examine the model's generation performance. Based on the analysis results in Appendix A.3 and Figure 7, we found that ESGM still maintains strong object mask extraction and generation capabilities even under severe occlusion conditions. However, when the generated mask shape exhibits certain anomalies, it does indeed produce objects matching that distorted shape. Nevertheless, this does not affect the overall FID and trainability of the generated images. While the shapes we currently extract may exhibit edge anomalies in the object mask due to occlusion issues, complete errors are extremely rare. Despite this, we can explore solutions for addressing erroneous shapes in future research.

---

> ### Author Response · Authors · 2025-11-21
> **Official Comment by Authors**
>
> > **Weakness4: The model is only evaluated on two datasets (DIOR-R and DOTA), both of which are airborne/satellite optical imagery.**
>
> This paper focuses on the field of remote sensing, hence all datasets employed belong to this domain. The selection of the DIOR-R and DOTA datasets is primarily based on two reasons. First, these two datasets are widely recognized as the most classic datasets in the field of remote sensing image object detection, as referenced in ([1] Gui, S., Song, S., Qin, R., & Tang, Y. (2024). Remote sensing object detection in the deep learning era—a review. Remote Sensing, 16(2), 327. [2] Wang, Y., Bashir, S. M. A., Khan, M., Ullah, Q., Wang, R., Song, Y., ... & Niu, Y. (2022). Remote sensing image super-resolution and object detection: Benchmark and state of the art. Expert Systems with Applications, 197, 116793. Second, other representative methods such as Aerogen and CC-Diff also utilize these two datasets in the remote sensing domain. Therefore, we adopt these two datasets as well. Additionally, we incorporate the HRSC-2016 dataset for model performance evaluation to validate the model's effectiveness in remote sensing. This dataset was selected due to its representativeness in remote sensing object detection and its adoption by other methods for validation ([3] Tang, D., Cao, X., Wu, X., Li, J., Yao, J., Bai, X., ... & Meng, D. (2025). AeroGen: Enhancing remote sensing object detection with diffusion-driven data generation. In Proceedings of the Computer Vision and Pattern Recognition Conference (pp. 3614-3624).). Results based on the HRSC2016 dataset are presented in the Appendix A.4.

---

> ### Comment · Reviewer_HT6t · 2025-11-26
> **Thank you for the rebuttal**
>
> I would like to thank the authors for their thoughtful responses to my questions and concerns. I will maintain my current positive scoring of the paper since the authors did respond to my questions sufficiently.

---

> > ### Author Response · Authors · 2025-11-26
> > **Official Comment by Authors**
> >
> > Dear Reviewer HT6t,
> >
> > Thank you very much for your thoughtful feedback and for maintaining your positive assessment of our work. We greatly appreciate your support.
> >
> > Best regards,
> >
> > Authors

---

### Official Review · Reviewer_3Som · 2025-11-01

**Soundness:** 3
**Presentation:** 3
**Contribution:** 2
**Rating:** 6
**Confidence:** 5

**Summary:**

This paper introduces Object Fidelity Diffusion (OF-Diff), a novel layout-to-image diffusion model designed specifically for generating high-fidelity remote sensing (RS) imagery. The primary motivation is to address critical failure modes in existing methods, such as control leakage, structural distortion, and dense generation collapse, which limit their utility for downstream tasks like object detection.

**Strengths:**

1. ESGM: Leverages pre-trained vision-language and segmentation models (RemoteCLIP and RemoteSAM) to extract precise object shape masks, providing strong geometric priors beyond simple bounding boxes.

2. Employs a teacher-student architecture where a "teacher" decoder (conditioned on both image and shape features) guides a "student" decoder (conditioned only on shape features). This allows the model to learn to generate high-fidelity textures and details without requiring real image references during inference.

3. Applies Denoising Diffusion Policy Optimization (DDPO) as a post-training step, using a reward function based on KNN distance and KL divergence to enhance the diversity and distributional consistency of the generated images.

Comprehensive experiments on the DIOR and DOTA datasets demonstrate that OF-Diff outperforms state-of-the-art methods in generation fidelity, layout consistency, and its utility in improving downstream object detection performance.

**Weaknesses:**

1. The ESGM module is critically dependent on two large, specialized models: RemoteCLIP and RemoteSAM. While effective, this raises questions about the framework's scalability, accessibility, and potential biases inherited from these foundational models. The paper could benefit from a discussion on the computational cost of this "template extraction" phase and an analysis of how errors from ESGM might propagate through the diffusion pipeline.

2. The paper clearly defines the DDPO reward function (Equation 9) but omits crucial implementation details for the KNN component. As the authors know, computing KNN in the high-dimensional pixel space is infeasible and perceptually meaningless. It is standard practice to compute this in a low-dimensional embedding space (e.g., using a CLIP or VAE encoder). While this is likely what the authors did, this critical detail should be explicitly stated in the implementation section to ensure reproducibility and clarity. Mentioning the specific pre-trained encoder used would be essential.

3. The paper exclusively focuses on object detection as the downstream task. While this is a highly relevant application, remote sensing involves many other perception tasks, such as semantic segmentation and change detection. Demonstrating the utility of OF-Diff for these other tasks could further strengthen the paper's claims of general applicability.

**Questions:**

Robustness of ESGM: The quality of the generated images seems highly dependent on the quality of the shape masks produced by ESGM. How does OF-Diff perform when ESGM fails or produces a distorted mask (e.g., for objects with complex boundaries or under heavy occlusion)? Is there a mechanism to handle such failures, or does the model simply replicate the distorted shape?

On the DDPO Reward Function: The KNN reward term encourages diversity by pushing generated samples away from the nearest neighbors in the real dataset. Could this potentially penalize the generation of "typical" or common instances and favor only rare or outlier-like objects? How was the balance between the KNN and KL terms (controlled by $\omega$) determined to prevent this?

Inference Speed and Cost: Could you provide details on the inference speed of OF-Diff compared to other methods? Specifically, since ESGM is only used to populate a mask pool for inference, how large does this pool need to be for good performance, and does the selection from this pool add any significant overhead?

Regarding the use of captions (Section 4.5): You note that including captions improves aesthetic appeal but harms downstream performance by deviating from the real data distribution. This is a very interesting finding. Does this imply that for data augmentation purposes, it is better to have models that are "faithful" to the original dataset's quirks and potential imperfections rather than models that generate more "idealized" or aesthetically pleasing images? I would appreciate it if you could elaborate on this insight.

---

> ### Author Response · Authors · 2025-11-21
> **Official Comment by Authors**
>
> Thank you very much for taking the time to review and for your support. We try our best to address your questions as follows.
>
> > **Weakness1 & Question1: The ESGM module is critically dependent on two large, specialized models: RemoteCLIP and RemoteSAM. While effective, this raises questions about the framework's scalability, accessibility, and potential biases inherited from these foundational models. The paper could benefit from a discussion on the computational cost of this "template extraction" phase and an analysis of how errors from ESGM might propagate through the diffusion pipeline. Robustness of ESGM: The quality of the generated images seems highly dependent on the quality of the shape masks produced by ESGM. How does OF-Diff perform when ESGM fails or produces a distorted mask (e.g., for objects with complex boundaries or under heavy occlusion)? Is there a mechanism to handle such failures, or does the model simply replicate the distorted shape?**
>
> Thank you for this important concern. We acknowledge that the model may inherit potential biases from these foundational models. We assess the model's robustness by analyzing the results generated from shape masks of varying quality.
>
> First, we discuss computational cost. When aiming to improve downstream task performance, RemoteCLIP is unnecessary. We investigated the fidelity, aesthetic quality, and trainability of model-generated images with and without captions; detailed procedures and analysis are provided in the Appendix A.7 and Figure 9. Experiments demonstrate that while images without captions are perceived as less aesthetically pleasing by humans, they yield higher performance gains in downstream tasks. Their FID value of 29.92 is significantly better than 42.17. Additionally, RemoteSAM's computational cost is acceptable. We tested improved methods like SAM and RSPrompter. Experiments confirm RemoteSAM holds significant advantages in inference speed and mask generation, with a generation speed of 0.3 seconds per mask being entirely acceptable.
>
> Second, we analyze the impact of a distorted mask on the model's generated results. Specifically, we selected cases such as objects under heavy occlusion to examine the model's generation performance. Based on the analysis results in Appendix A.3 and Figure 7, we found that even under severe occlusion conditions, ESGM still demonstrates strong object mask extraction and generation capabilities. However, when the generated mask shape exhibits certain anomalies, it does produce objects matching that distorted shape. Nevertheless, this does not affect the overall FID and trainability of the generated images. Although the shapes we currently extract may exhibit edge anomalies in the object mask due to occlusion and other issues, complete errors are extremely rare. Nevertheless, we can explore solutions for addressing erroneous shapes in future research.
>
> > **Weakness2: The paper clearly defines the DDPO reward function (Equation 9) but omits crucial implementation details for the** **KNN** **component. As the authors know, computing KNN in the high-dimensional** **pixel** **space is infeasible and perceptually meaningless. It is standard practice to compute this in a low-dimensional embedding space (e.g., using a CLIP or** **VAE** **encoder). While this is likely what the authors did, this critical detail should be explicitly stated in the implementation section to ensure reproducibility and clarity. Mentioning the specific pre-trained encoder used would be essential.**
>
> Thank you for this valuable suggestion. As you mentioned, we do indeed compute this in a low-dimensional embedding space using the CLIP image encoder. We have also incorporated your feedback by adding detailed implementation specifics to the Appendix A.2 of the paper.
>
> > **Weakness3: The paper exclusively focuses on object detection as the downstream task. While this is a highly relevant application,** **remote sensing** **involves many other perception tasks, such as semantic segmentation and change detection. Demonstrating the utility of OF-Diff for these other tasks could further strengthen the paper's claims of general applicability.**
>
> Thank you for this valuable suggestion. We will conduct further testing on other downstream remote sensing tasks such as semantic segmentation and change detection. Considering that performance testing for other downstream tasks requires acquiring new datasets and adding annotation information for segmentation, change detection, and other processes, this involves significant time costs and is difficult to complete in the short term. We will take your advice into consideration in the future and validate the model's practicality for other downstream tasks.

---

> ### Author Response · Authors · 2025-11-21
> **Official Comment by Authors**
>
> > **Question2: On the DDPO Reward Function: The** **KNN** **reward term encourages diversity by pushing generated samples away from the nearest neighbors in the real dataset. Could this potentially penalize the generation of "typical" or common instances and favor only rare or outlier-like objects? How was the balance between the KNN and KL terms (controlled by) determined to prevent this?**
>
> Thank you for this valuable suggestion. We have conducted experiments to analyze the balance between KL and KNN.
>
> Using only the KNN reward does indeed promote diversity at the cost of generating excessive outliers, which is what you referred to as neglecting typical or common instances. Therefore, we need the KL term to strike a balance. This ensures generated samples maintain high fidelity while achieving sufficient diversity to enhance trainability. We tested multiple combinations of KNN and KL weight ω to assess their impact on trainability. Experimental results indicate that ω=2 achieves the optimal balance between the two.
>
> | ω    | Trainability |
> | ---- | ------------ |
> | 0.25 | -7.83%       |
> | 0.5  | -2.74%       |
> | 1    | +0.83%       |
> | 2    | +1.47%       |
> | 4    | +0.51%       |
>
> > **Question3: Inference Speed and Cost: Could you provide details on the inference speed of OF-Diff compared to other methods? Specifically, since ESGM is only used to populate a mask pool for inference, how large does this pool need to be for good performance, and does the selection from this pool add any significant overhead?**
>
> Thank you for this valuable suggestion. We have provided details on the inference speed of OF-Diff compared to other methods here and in the Appendix. Since our pool uses only 200 shapes, there is virtually no additional overhead.
>
> |   models   |  FID  | train GPU mean memory | train GPU hours | inference meantime/sample (s) |
> | :--------: | :---: | :-------------------: | :-------------: | :---------------------------: |
> | Layoutdiff | 37.60 |         29232         |      41.33      |             ＜1s              |
> |   GLEGEN   | 35.06 |         14186         |      57.76      |             5.18              |
> |  aerogen   | 27.78 |         27634         |      49.52      |             1.85              |
> |  cc-diff   | 49.62 |         13668         |      38.01      |             3.96              |
> |  of-diff   | 24.92 |         27340         |      44.27      |             3.42              |
>
> > **Question4: Regarding the use of captions (Section 4.5): You note that including captions improves aesthetic appeal but harms downstream performance by deviating from the real data distribution. This is a very interesting finding. Does this imply that for data augmentation purposes, it is better to have models that are "faithful" to the original dataset's quirks and potential imperfections rather than models that generate more "idealized" or aesthetically pleasing images? I would appreciate it if you could elaborate on this insight.**
>
> Thank you for this important concern. This is an interesting question.
>
> First, we conducted experiments on whether to use captions as a variable. Results showed that using captions improved mAP by 0.52%, while not using captions improved it by 1.47%. Additionally, the FID value for the non-caption approach was 24.92, lower than the 42.17 achieved with captions.
>
> Second, Reviewer HT6t also raised questions about adding captions. Following their suggestion, we conducted a user study. We surveyed 14 human remote sensing/AI experts, obtaining their evaluations of the two generated images in terms of “better alignment with real-world remote sensing imagery” and “greater human visual preference.” Details are provided in the Appendix A.7.
>
> Third, we performed feature extraction on both generated images (with and without captions) and real images, then visualized them in a two-dimensional space using t-SNE dimensionality reduction. This allows for a more intuitive understanding of their respective characteristics. See Figure 9 in the Appendix A.7 for details.

---

> > ### Comment · Reviewer_3Som · 2025-11-28
> >
> > Thanks for the reponses. These responses solved my concerns. I keep the positive rating unchanged.

---

> > > ### Author Response · Authors · 2025-11-28
> > > **Official Comment by Authors**
> > >
> > > Dear Reviewer 3Som,
> > >
> > > Thank you very much for your thoughtful feedback and for maintaining your positive assessment of our work. We greatly appreciate your support.
> > >
> > > Best regards,
> > >
> > > Authors

---

### Author Response · Authors · 2025-11-25
**Official Comment by Authors**

We sincerely thank all reviewers for their thorough and constructive reviews. We have carefully addressed all comments with detailed responses and made corresponding revisions to the manuscript. Below, we summarize the major changes in the paper in sequential order (highlighted in blue) and their correspondence to the reviewers' comments.

1. Line 268 and 741:
   - Adding implementation details for KNN and KL. (3Som)
2. Line 278, 349 and 793:
   - Adding the HRSC2016 dataset, we extend evaluation to a further high-resolution fine-grained benchmark. (HT6t, 8kyc)
3. Line 431, 471 and 850:
   - Adding a user study to clarify the impact of using captions. (HT6t)
4. Line 769 and 784:
   - Adding the robustness analysis when masks are heavily occluded. (HT6t, 7tpr)
5. Line 810 and 832:
   - Adding mAP results at different real-to-generated image training ratios. (8kyc)
6. Line 821 and 843:
   - Adding inference speed and computational cost. (3Som, 7tpr)
7. Line 922:
   - Adding experiments analyzing the aesthetic–downstream performance conflict. (3Som)
   - Adding experiments on the impact of data that deviate from the real data distribution. (8kyc)

---

### Author Response · Authors · 2025-11-26

Dear Reviewers,

We hope this message finds you well. We wanted to follow up on the rebuttal we submitted for our ICLR submission.

We know the review period keeps you busy, and we really appreciate the time you've put into reviewing our paper. We've worked hard to address the concerns from your review. Whenever you have a moment, we'd love to hear what you think about our responses.

If you need us to clarify anything else, just let us know. Thanks so much for your time.

Best regards,

Authors

---

### Comment · Area_Chair_tqRJ · 2025-11-26

Dear reviewers,

Please check the author's reply. Feel free to raise any questions or start a discussion, regardless of whether you will change the score.

Your AC.

---

### Author Response · Authors · 2025-12-01
**Thank AC for taking over our submission and Summary of Review-Rebuttal phase**

Dear Area Chair,

Thank you for handling our paper and for joining the dialogue with the reviewers. We recognize that this year's exceptional circumstances have imposed an extra workload on ACs, and we are truly grateful for your dedication. **To facilitate a rapid appraisal of our submission, we supply a succinct overview of (I) our contribution, (II) the reviewers' comments, and (III) the principal enhancements introduced in this rebuttal.**

 **(I) Our contribution**

This paper introduces Object Fidelity Diffusion (OF-Diff), which effectively addresses issues such as object structural distortion and dense generation collapse in controllable image generation for remote-sensing imagery.  OF-Diff leverages object layouts to extract structural shape priors and employs an online-distillation strategy to integrate complex image features. This allows the model to perform highly controllable, high-fidelity image generation at inference without relying on real-image references.      Furthermore, we introduce DDPO based on KNN and KL to fine-tune the diffusion process, making the generated remote sensing images more diverse and semantically consistent. Extensive experiments demonstrate the superior performance and trainability of OF-Diff for controllable RS image generation.

 **(II) Strengths recognized by the reviewers**

The reviewers praised our work for:

**(1) Novel method insights**: Reviewers 3Som, HT6t, and 8kyc expressed their approval of the pipeline’s feasibility and the effectiveness of the proposed methods;

**(2) Significant practical improvement**: OF-Diff operates at inference without real-image references, while still enabling controllable generation of objects with accurate shape characteristics and complex multi-object scenes. (HT6t, 8kyc)

**(3) Strong empirical results**: Comprehensive experiments demonstrate that OF-Diff outperforms SOTA methods in generation fidelity, layout consistency, and its utility in improving downstream object detection performance. (3Som, HT6t, 8kyc)

 **(III) Key improvements made during rebuttal**

**(1) Incorporate additional datasets and conduct more comprehensive experiments.**

- Adding the HRSC2016 dataset, we extend evaluation to a further high-resolution fine-grained benchmark. (HT6t, 8kyc)
- Adding mAP results at different real-to-generated image training ratios. (8kyc)
- Adding ablations on KNN–KL weight ω to assess the impact on trainability. (3Som, 8kyc)

**(2) In-Depth Analysis and user study**

- Adding experiments analyzing the aesthetic–downstream performance conflict. (3Som)
- Adding the robustness analysis when masks are heavily occluded. (HT6t, 7tpr)

- Adding a user study to clarify the impact of using captions. (HT6t)

**(3) Supplementary Experimental Details**

- Adding implementation details for KNN and KL. (3Som)

- Adding inference speed and computational cost. (3Som, 7tpr)

 **(IV) Unanimous Reviewer Endorsement**

**The reviewers ultimately expressed a uniformly positive stance toward our paper, confirming that our improvements have fully addressed their concerns:**

- **Reviewer 3Som (original rating: 6, after rebuttal: 6):** "These responses solved my concerns. I keep the positive rating unchanged."

- **Reviewer HT6t (original rating: 6, after rebuttal: 6):** "I would like to thank the authors for their thoughtful responses to my questions and concerns. I will maintain my current positive scoring of the paper since the authors did respond to my questions sufficiently."
- **Reviewer 8kyc (original rating: 6, after rebuttal: 6)**
- **Reviewer 7tpr (original rating: 4, after rebuttal: the score already cannot be modified):** "Thank you for your reply. l have no additional problem. However, I found that the score cannot modified."



We hope this summary facilitates your evaluation and remain available for any clarifications.

Best regards,

The Authors

---

### Meta-Review · Area_Chair_x8tf · 2026-01-07

**Summary:**

The paper received three marginal accept ratings (6) and one lower score (4). The main concerns were: (i) whether the contribution is sufficiently novel vs. a combination of existing components, (ii) heavy reliance on RemoteCLIP/RemoteSAM and missing reproducibility details (e.g., KNN in the DDPO reward), (iii) lack of reported efficiency/cost, (iv) unclear robustness to noisy/occluded masks, and (v) limited evidence on generalization / OOD augmentation and how much synthetic data actually helps downstream detection. After rebuttal, most reviewers indicated their concerns were addressed and kept their positive stance, so the AC leans borderline accept.

**Reviewer Concerns:**

Addressed by rebuttal:
(1) Added missing implementation details for the KNN/KL reward and provided additional analyses/experiments; Reviewer 3Som explicitly said the responses solved their concerns.
(2) Reported computational cost and expanded analysis of the caption trade-off and robustness to occlusion/noisy masks (responding directly to HT6t’s listed weaknesses).
(3) Added experiments on trainability vs. real/synthetic ratios (requested by 8kyc).

Still outstanding:
(1) The novelty question remains somewhat subjective: Reviewer 7tpr still flagged novelty as a possible acceptance-threshold issue, even though most other concerns were addressed.
(2) Stronger cross-dataset/OOD augmentation validation remains limited (explicitly requested by 8kyc).

**Reviewer Scores:**

Reviewer 3Som: likely unchanged (6 → 6); they stated the rebuttal solved their concerns and kept the rating.

Reviewer HT6t: likely unchanged (6 → 6); they indicated they would maintain the positive score after the rebuttal addressed their questions.

Reviewer 8kyc: likely unchanged (6 → 6) (original and after rebuttal both 6).

Reviewer 7tpr: likely unchanged (4) or increase to 6; they wrote they had “no additional problem” after the rebuttal, but could not modify the score, while earlier still noting novelty concerns.

---

### Decision · Program_Chairs · 2026-01-26

Accept (Poster)